# Non-native ants are breaking down biogeographic boundaries and homogenizing community assemblages

Lucie Aulus-Giacosa [1] ✉, Sébastien Ollier[1,2] & Cleo Bertelsmeier [1] ✉

As geographic distance increases, species assemblages become more distinct, defining global biogeographic realms with abrupt biogeographic boundaries. Yet, it remains largely unknown to what extent these realms may change because of human-mediated dispersal of species. Focusing on the distributions of 309 non-native ant species, we show that historical biogeographic patterns have already broken down into tropical versus non-tropical regions. Importantly, we demonstrate that these profound changes are not limited to the distribution patterns of non-native ants but fundamentally alter biogeographic boundaries of all ant biodiversity (13,774 species). In total, 52% of ant assemblages have become more similar, supporting a global trend of biotic homogenization. Strikingly, this trend was strongest on islands and in the tropics, which harbor some of the most vulnerable ecosystems. Overall, we show that the pervasive anthropogenic impacts on biodiversity override biogeographic patterns resulting from millions of years of evolution, and disproportionally affect particular regions.

Human mobility and trade have exploded in the Anthropocene, causing voluntary and accidental dispersal of thousands of species worldwide[1–4]. Some of these species have been able to establish outside of their native range (hereafter referred to as non-native species)[5]. The number of emergent non-native species[6] and their range sizes are predicted to increase even further[7–9], changing the composition of species assemblages worldwide. Historically, the spatial turnover patterns in species assemblages (β diversity) were characterized by several abrupt transitions, called "biogeographic boundaries". One famous example of a biogeographic boundary is the Wallace line separating the Indomalayan and the Australasian realms. Biogeographic boundaries have been shaped by geography, past and present environmental differences and evolutionary history[10,11]. However, the reshuffling of biodiversity with human-mediated transport has the potential to break these historical biogeographic boundaries[11,12]. Previous studies on terrestrial gastropods[13], reptiles, and amphibians[14] have focused on non-native species distributions in their native and current ranges (i.e., before and after human-mediated dispersal) and found a reduction in the number of distinct bioregions. Moreover,

recent research on vertebrates has shown that human-mediated introductions and species extinctions alter biogeographic boundaries, with marked differences according to the taxonomic group[15]. However, it remains unclear to what degree this occurs in insects, which outnumber all other known animal species[16], and if more subtle changes in biogeographic boundaries may be revealed with more extensive spatial coverage. More importantly, it is still an open question if, and to what extent, non-native species dispersal affects the biogeographic boundaries of biodiversity in general, including all native species within a taxonomic group. Answering these questions is crucial to understand to what extent the globalization of trade and transport is leading to a globalization of species assemblages.

In parallel with these changes in biogeographic boundaries, the global movement of species may either lead to the homogenization or differentiation of species assemblages. Homogenization may happen if the same set of species is introduced in several regions, which become increasingly similar in terms of species composition as a result. Alternatively, differentiation of assemblages[17] could happen due to invasions of different non-native species. Recently, biotic homogenization

[1]Department of Ecology and Evolution, Biophore, UNIL – Sorge, University of Lausanne, 1015 Lausanne, Switzerland. [2]Université Paris – Saclay, CNRS, AgroParisTech, Ecologie Systématique Evolution, 91405 Orsay, France. ✉e-mail: lucie.aulus@unil.ch; cleo.bertelsmeier@unil.ch

has become a major topic in global change ecology, with numerous local and regional studies, including plants in China[18,19], micro-crustacean communities in Brazil[20], and island birds[21]. Yet, most previous studies have a limited geographic and taxonomic focus[20,22] and almost exclusively investigate the biogeography of non-native species assemblages[13], despite the obvious importance of assessing consequences for native species as well[23]. Importantly, these previous studies have measured "homogenization" over the whole extent of a study region, while in fact, there may be large regional differences, with some areas becoming, on average, more similar to all other regions, and some becoming increasingly dissimilar. For example, the impact of non-native species is expected to be stronger on islands than on the mainland[24] because islands have lower native species richness[25,26]. Additionally, islands harbor high numbers of evolutionary unique, and geographically restricted species making them more vulnerable to human impacts[24,27]. More generally, it is unknown to what extent different parts of the world are being homogenized or are differentiating at different rates.

To address these questions, we used ants (Formicidae) as a model system. Ants dominate terrestrial ecosystems in terms of their abundance (20 x 10^15 ground-dwelling individuals) and biomass[28], they occupy various trophic positions[29,30] and are present in nearly all terrestrial habitats in every continent[31,32]. Ants are key contributors to many ecosystem functions, such as seed dispersal[33], soil bioturbation[34], resource removal[35], pest control, and help structure most invertebrate communities through predation or competition[36]. Ants are also prominent as non-native species, with at least 309 species established outside of their native range, and 17 being listed as highly problematic[37]. Many of these non-native ants displace native species, altering community structure and impairing ecosystem functions[38], and cause estimated mean annual economic costs of approximately 398 million US$ globally[38,39]. Moreover, ants are a good model system for studying unintentional species introductions. Unlike many other taxa, no ant invasions are thought to have resulted from the deliberate introduction of species as pets, ornamentals, or biocontrol agents[40].

Here, our aim is to test to what extent non-native ant species dispersal changes biogeographical boundaries. Specifically, we test the hypothesis that a general trend toward biotic homogenization is accompanied by large regional differences, with stronger homogenization on islands (due to their depauperate faunal composition and greater vulnerability to invasion[25]) and tropical areas (since they are climatically similar to the native ranges of most non-native ant species). Ant species distributions have recently been mapped across 536 countries and sub-country spatial entities (hereafter polygons)[31,32,41] and global ant biodiversity (known and undiscovered) has been recently mapped at an even finer grain[42]. Here, we separately analyzed biogeographic patterns before and after human-mediated dispersal for assemblages of non-native ant species (309 species) and all ant species with known distribution records (13,774 species, hereafter referred to as "all ant species") at the polygon level to explore the reshaping of biogeographic boundaries and biotic homogenization due to the global movement of non-native ant species. We conducted the analyses at the global level, and then compared the patterns on the mainland and islands separately (Fig. 1). We show that the pervasive anthropogenic impacts on biodiversity can override historical biogeographic patterns, and that biotic homogenization can be heterogeneous in space and vary in intensity. Moreover, we identify tropical islands as especially vulnerable to homogenization.

## Results
### Global biogeographic realms and boundaries
We found five biogeographic realms (hereafter realms) of ants before human-mediated dispersal (Fig. 2), based on a hierarchical clustering analysis on the pairwise compositional dissimilarity ($\beta_{sim}$) of assemblages using the unweighted pair group method with arithmetic mean

(UPGMA, see Methods and Fig. 1). These mostly correspond to Wallace's classical realms: Nearctic, Neotropical, Palearctic, Ethiopian, and Oriental-Australian realms[11] (Fig. 2c). The native ranges of 309 non-native ant species are representative of this general biogeographic pattern, since they also clustered into five realms (Fig. 2a), which coincided to a large extent with the five major realms that were delineated for all ant species (Fig. 2c), at the exception of the Nearctic realm which is grouped with the Neotropical realm, and New Zealand that is separate from the Oriental-Australian realm.

To test if the dispersal of non-native species has reshaped these historical biogeographic patterns, we analyzed the changes in compositional dissimilarity of non-native ant assemblages before and after human-mediated dispersal (i.e., including both the species' native and non-native ranges). We found a reduction from five to four realms for non-native ant species (Fig. 2b), with a large pantropical cluster and three non-tropical clusters which correspond to the Nearctic, Eastern Palearctic, and Western Palearctic realms. These results concur with the findings for terrestrial gastropods[13] where human-mediated transport also resulted in the formation of a single tropical cluster and a temperate one. This reduction in delineated realms among species assemblages is, to some extent, expected given the large-scale movement of non-native species around the planet. However, only 2.2% of ant species used in this study (13,774) have been introduced outside their native range, and it is still unclear if this has the power to redefine biogeographic realms for all ant species. To test this, we analyzed the composition of species assemblages containing all ants (13,774 species), including the non-native range of non-native ant species. Surprisingly, we found that the global dispersal of a relatively small number of non-native species resulted in a remarkable change in biogeographic realms (Fig. 2d). After human-mediated dispersal, there is a new pantropical realm mainly composed of the former Ethiopian, Neotropical, Oriental, and Australian realms and four other realms consisting of two new realms (India and Southern Neotropics) and the former Nearctic and Palearctic realms. The effect of non-native species dispersal was more pronounced in the tropics, likely due to a higher number of non-native species both originating from the tropical mainland (GLMM, $p < 0.01$) and having been introduced within the tropics (GLMM, $p < 0.0001$, Supplementary Fig. 2).

### Greater biogeographic changes on islands
To test to what extent the changes in biogeographic patterns are driven by island versus mainland assemblages, we did separate clustering analyses on the pairwise compositional dissimilarity of islands and mainlands for all ant species (Fig. 3 and Supplementary Fig. 5 for non-native ants). Before human-mediated dispersal, island assemblages (Fig. 3c) fell within the same realms as the adjacent mainland assemblages (Fig. 3a). In total, there were seven realms, with slight differences between mainlands and islands. Notably, the Australian mainland realm was divided into an Oriental-Oceanian realm and southern Australian islands.

After human-mediated dispersal of non-native species, the number of biogeographic realms decreased for both mainland and islands, with respectively five and four remaining realms (Fig. 3b–d). Strikingly, most ant assemblages on tropical islands form a single new realm spanning the whole circumference of the planet (Fig. 3d), similar to the pattern in our global dataset (Fig. 2). However, when we analyzed mainland assemblages separately from islands, the effect of non-native species dispersal was much weaker. This suggests that non-native ant species dispersal had a much greater impact on island biogeography than on the mainland. A possible explanation is that the ratio of non-native to native species is greater on islands ($0.94 \pm 1.73$) than on mainlands ($0.05 \pm 0.14$) (Wilcoxon test, $p < 0.0001$, Supplementary Fig. 3).

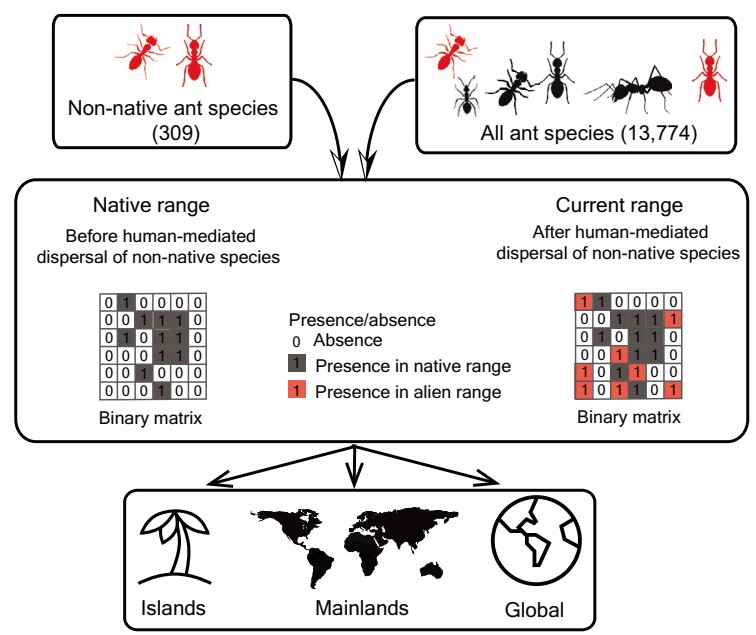

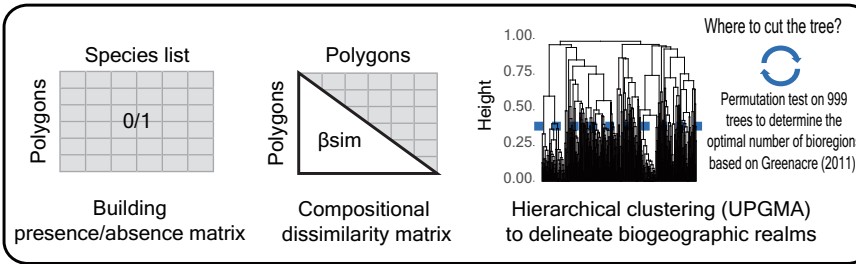

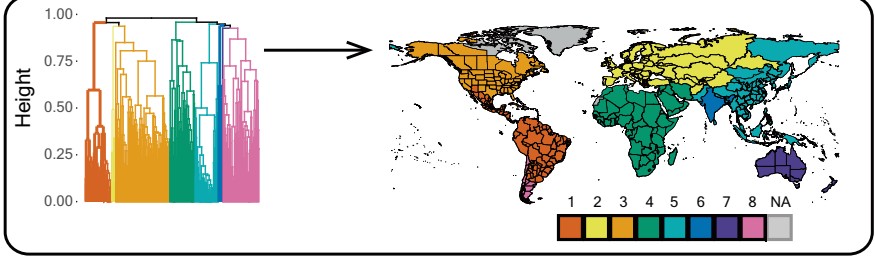

**Fig. 1 | Conceptual diagram of analysis steps, data flow, and biogeographical analysis.**

## Homogenization of ant species assemblages

To measure the degree of biotic homogenization in ant assemblages after human-mediated dispersal, we calculated a homogenization index ($h_{i,j}$) as the change in compositional dissimilarity ($\beta_{sim}$) across all pairwise comparisons of polygons and $\bar{h}_{i\bullet}$ the average value of the homogenization index by polygon (see Methods).

Globally 52% of pairs of ant assemblages have become more similar to each other (i.e., have been homogenized), while 7% have become more dissimilar (i.e., have differentiated). Moreover, we found that the degree of homogenization differs among regions, with island assemblages homogenizing more than mainland assemblages (61 and 48% of ant assemblages subjected to biotic homogenization, respectively). To test if the average degree of assemblage homogenization ($\bar{h}_{i\bullet}$) was linked to the location on islands and/or within the tropics, we used a non-parametric two-way ANOVA (see Methods). The most homogenized assemblages were located on islands ($p < 0.01$, islands: $\bar{h}_{i\bullet} = -0.057 \pm 0.057$, mainlands: $\bar{h}_{i\bullet} = -0.023 \pm 0.021$), with more notable effects in the tropics ($p < 0.001$, tropical: $\bar{h}_{i\bullet} = -0.048 \pm 0.042$, non-tropical: $\bar{h}_{i\bullet} = -0.021 \pm 0.031$) (Fig. 4a). Strikingly, the most

homogenized assemblages also correspond to ant biodiversity hotspots such as the northern Neotropic, Ethiopian, Madagascan, Oriental and Australian regions[43].

To test if assemblages become more similar on average to other assemblages because they have received many non-native species (recipient regions), or because they have many species which have established non-native populations elsewhere (donor regions), we used negative binomial generalized linear mixed models (GLMMs, see Methods). Tropical islands were greater recipient regions of non-native ant species, contributing to their biotic homogenization (Fig. 4c, Wilcoxon test, $p < 0.001$). In contrast, at a global level the homogenization index of mainlands decreases because mainlands are greater donor regions of non-native species (that often become established on tropical islands), thereby becoming more similar to the assemblages of the recipient regions. Consequently, when considering mainlands separately, there are no dramatic biogeographic changes given that the non-native species they receive represent a smaller fraction of all species than on islands (Supplementary Fig. 3) and the non-native species they have donated to other assemblages mostly

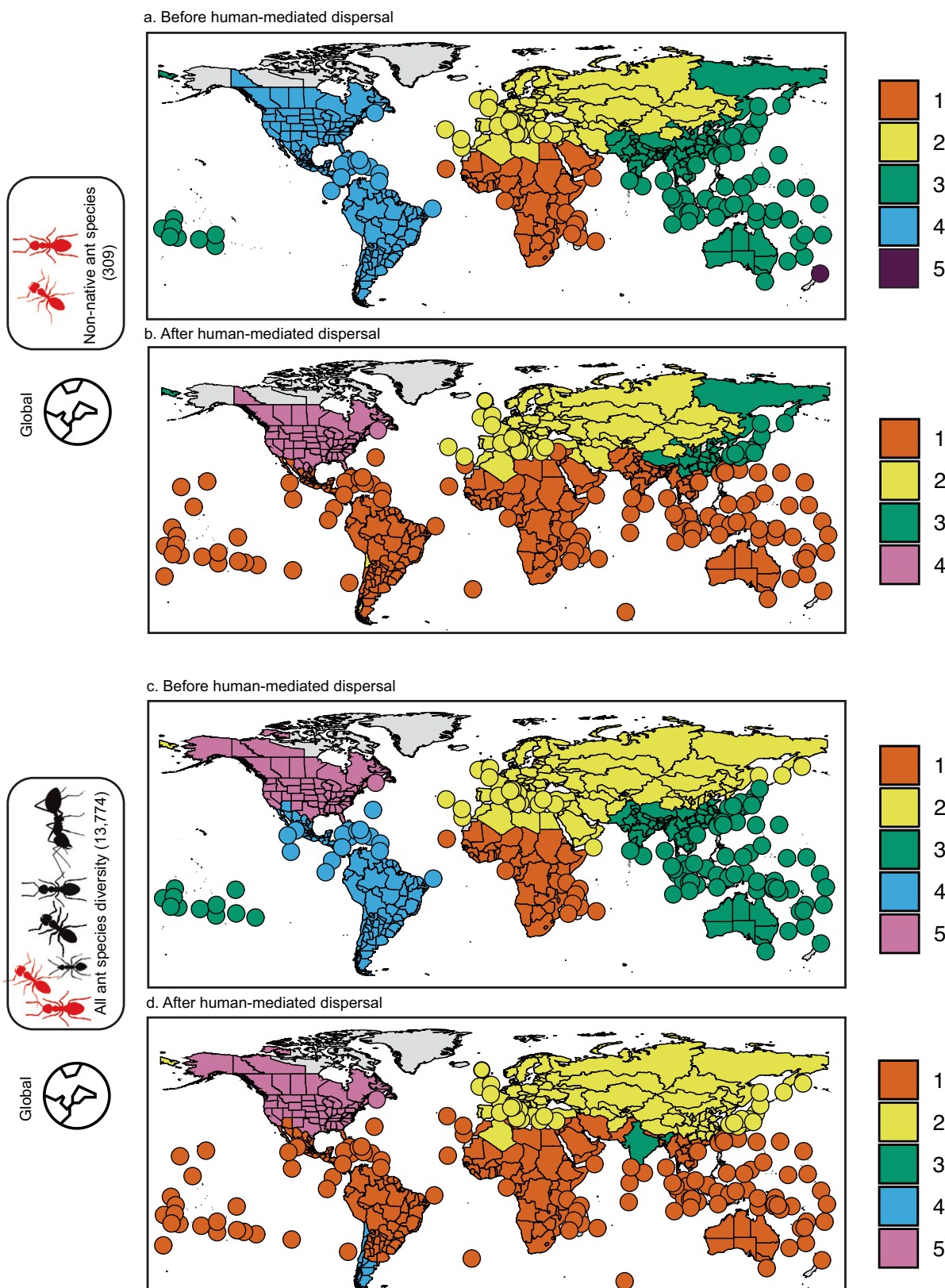

**Fig. 2 | Global biogeographic patterns before and after human-mediated dispersal of non-native species.** Biogeographic groups of 309 non-native ant species: **a** before (native ranges) and **b** after human-mediated dispersal (native + non-native ranges). Biogeographic realms of all ant species (13,774): **c** before (native ranges) and **d** after human-mediated dispersal of non-native species (native + non-native

ranges). Colors indicate biogeographic clusters identified using compositional dissimilarity ($\beta_{sim}$ index) and clustering analysis. Light gray areas highlight polygons for which no distributional data were available. Source data are provided as a Source Data file.

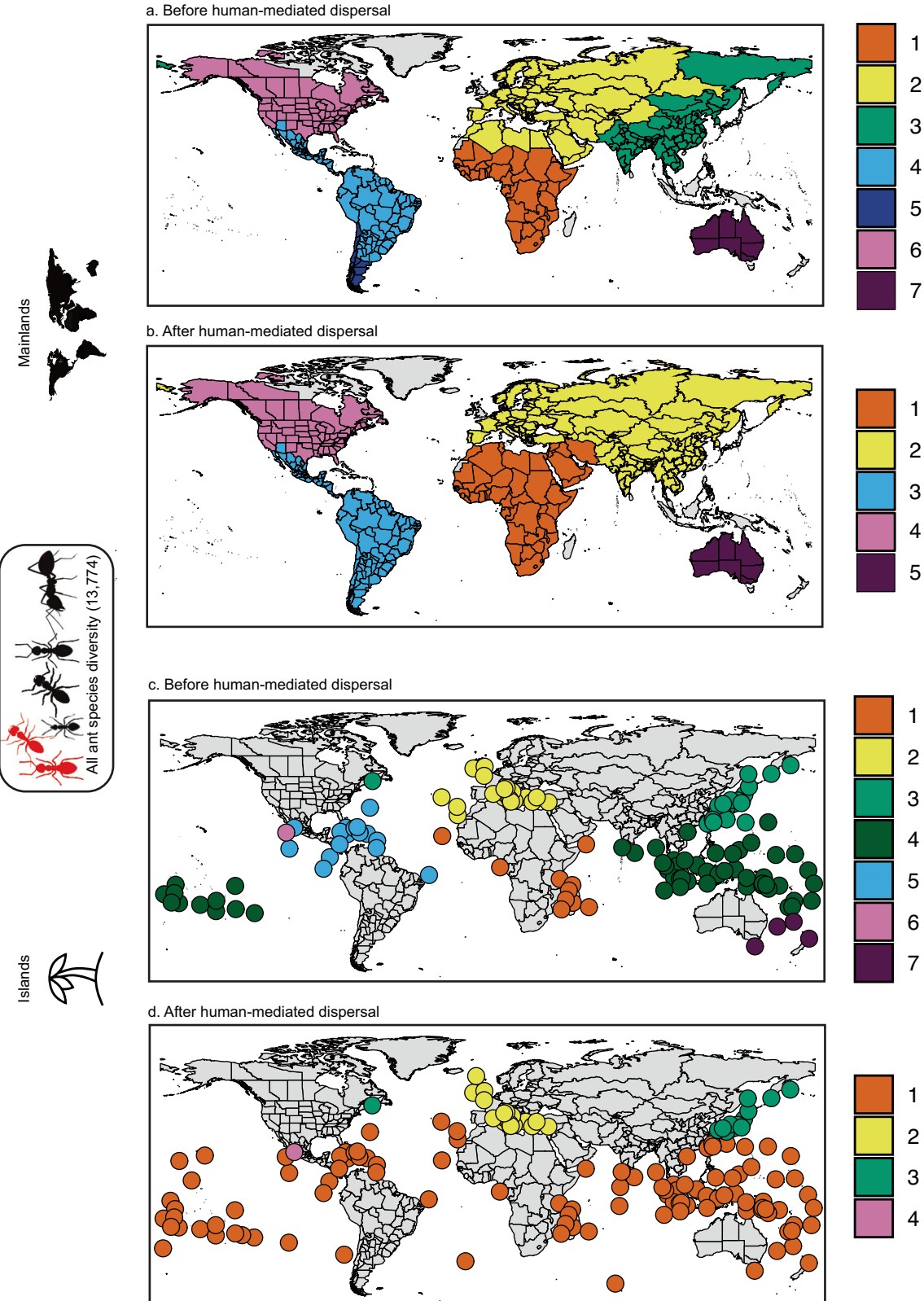

**Fig. 3 | Changes in biogeographic boundaries split by mainlands and islands.** Biogeographic realms of all ant species (13,774) on mainlands (top) and islands (bottom): **a**–**c** before (native ranges) and **b**–**d** after human-mediated dispersal (native + non-native ranges) of non-native species. Colors indicate realms identified using compositional dissimilarity ($\beta_{sim}$ index) and clustering analysis. Source data are provided as a Source Data file.

a. Distribution of homogenization values by categories

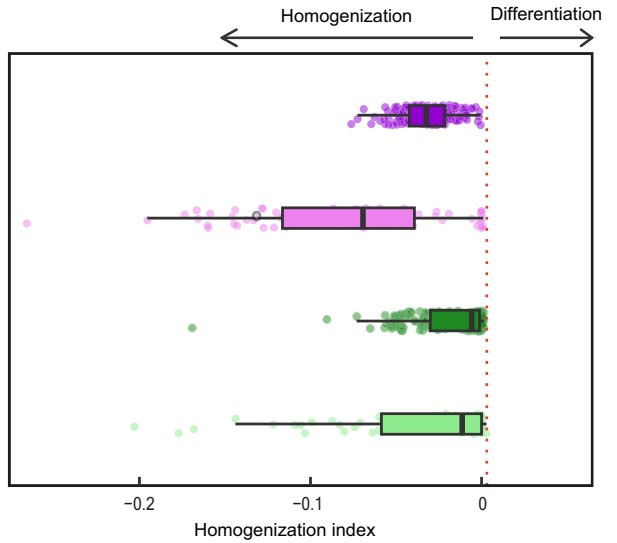

c. Number of non-native species donated and received by categories

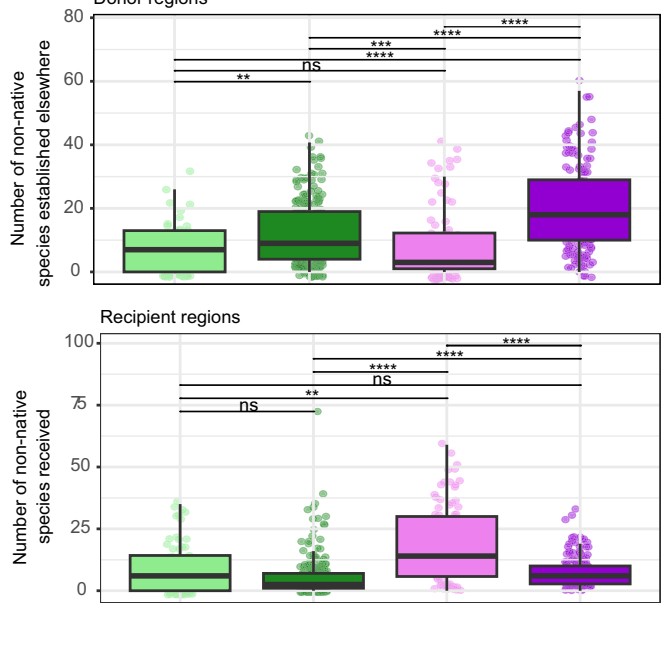

Categories

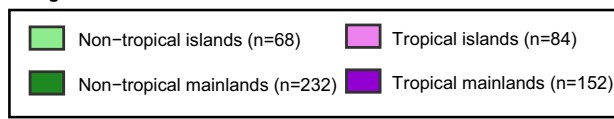

b. Degree of homogenization by polygons

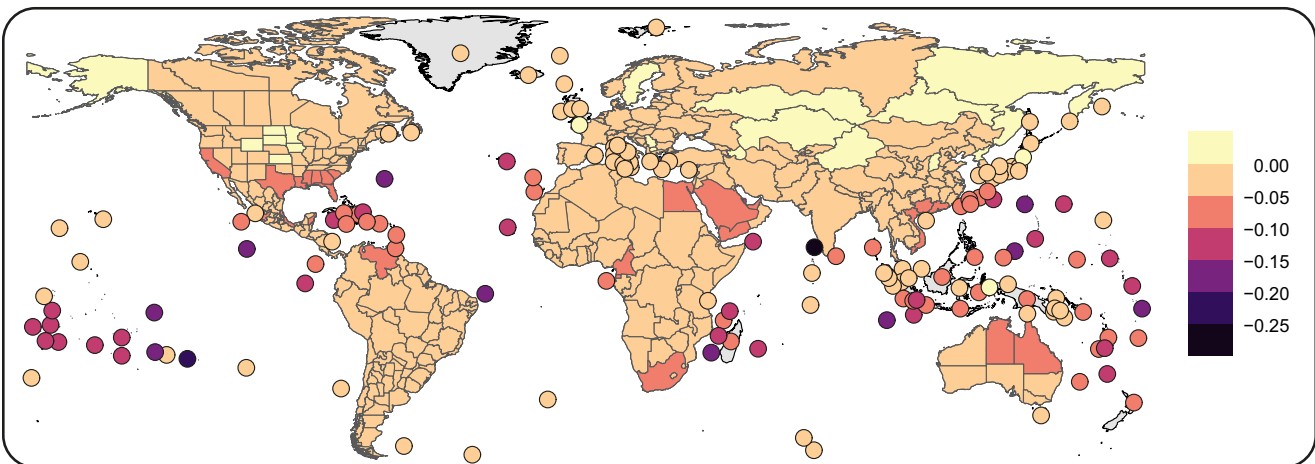

**Fig. 4 | Global homogenization trend for all ant species (13,774 species) at the global scale. a** Homogenization by polygon ($\bar{h}_{i\bullet}$) grouped by categories of climatic status (tropical vs. non-tropical) and geographic status (mainlands vs. islands). **b** Average degree of homogenization ($\bar{h}_{i\bullet}$) by polygon. **c** Number of donated and received non-native species grouped by categories of climatic status (tropical vs. non-tropical) and geographic status (mainlands vs. islands). Differences between groups were tested using a two-sided Wilcoxon signed-rank test with Bonferroni correction, and significance are given with asterisk (ns non-significant, *: $0.05 < p \leq 0.1$, **: $0.01 < p \leq 0.05$, ***: $p \leq 0.001$). Box plots (**a–c**) represent data

from $n$ = 536 polygons (68 non-tropical islands, 232 non-tropical mainlands, 84 tropical islands, and 152 tropical mainlands) where the lower bound of lower whisker shows the minimum value of the data that is within 1.5 times the inter-quartile range under the 25th percentile, lower bound of box shows the lower quartile, center of box shows the median, upper bound of box shows the upper quartile, and upper bound of upper whisker shows the maximum value of the data that is within 1.5 times the interquartile range over the 75th percentile. Source data are provided as a Source Data file.

establish on islands. This likely explains why we did not detect a large pantropical realm when considering mainlands only (Fig. 3b), contrasting with the global pattern (Fig. 2d). These results emphasize that the reshaping of biogeographic realms in the Anthropocene is not a simple numbers game where the most species-poor regions are the most affected. Indeed, the global homogenization process depends on both exports (by donor regions) and imports (by recipient regions) of non-native species.

Before human-mediated transport, species assemblages that are geographically closer tend to share more similar species. To test if this relationship has been affected by the increasing homogenization of community assemblages, we measured the distance-decay (relationship between geographical distance and number of shared species) before and after human-mediated dispersal of non-native species (see Methods). Our analysis confirmed that the increasing dissimilarity of ant assemblages as a function of geographical distance weakened in

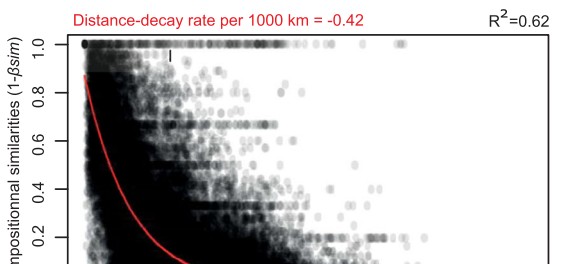

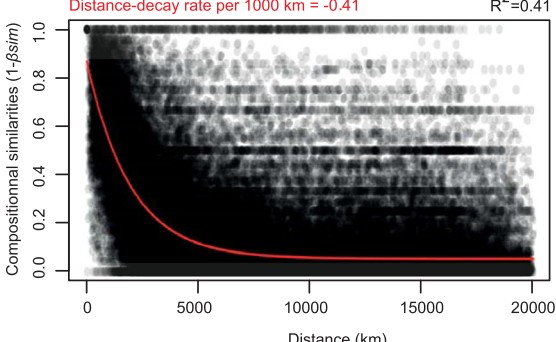

**Fig. 5 | Global distance-decay before and after human-mediated dispersal across all ant assemblages (13,774 species).** Distance-decay represents the relationship between compositional similarities ($1-\beta_{sim}$) of ant assemblages (for all ant species) and geographical distances between the centroids of polygons before the (**a**) and after (**b**) the human-mediated dispersal of 309 non-native ant species. Compositional similarities are fitted against distance with nls models. The R-square of the models and distance-decay rate per 1000 km are given above the graphs. Source data are provided as a Source data file.

the Anthropocene (Mantel r-statistics, 0.68 before and 0.37 after human-mediated dispersal of non-native species, $p < 0.001$, Fig. 5).

## Discussion

Our results show that the pervasive anthropogenic impacts on biodiversity redefine biogeographic patterns resulting from millions of years of evolution and natural dispersal, and disproportionally affect particular regions. Even though non-native species represent a small fraction of all ant species, they have already caused global homogenization of ant species assemblages. This is in line with the trend towards biotic homogenization found in other taxa and regions[13,18–22]. However, our study shows that such a profound impact not only on the biogeography of non-native species themselves[13,14], but for all species in a taxonomic group.

Moreover, we argue that it is crucial to move beyond the binary assessment of "homogenization" or "no homogenization" in a given study area, and to consider the complexities of species movements around the planet in greater detail. Here, our findings go beyond previous work and underline the importance of assessing regional heterogeneity, since many areas became more similar to other assemblages either because they were recipient or donor regions of non-native species. Moreover, some areas did not homogenize compared to other assemblages. Accounting for these aspects, we identified tropical regions and islands[25] as especially vulnerable, with the emergence of a new pantropical realm. This is particularly concerning as almost two-thirds of biodiversity hotspots[37] are located in tropical regions and islands are well-known centers of endemism[44,45]. We did not quantify the role of different environmental or socio-economic drivers of the observed changes, but as non-native ant species mostly originate from and are introduced in tropical areas (GLMM, $p < 0.0001$, Supplementary Fig. 2), climatic filtering is likely a main contributing factor in non-native species establishment[14]. Additionally, trade—and in particular the plant and fruit trade—is known to be an important introduction pathway of non-native ants[46] and could determine which locations within a suitable climatic area (the tropics) are more likely to be reached by non-native ants[13,14].

Our study presents several limitations. First, many ant species are not yet described[42], and our knowledge on the delimitation of species' native vs. non-native ranges is generally based on expert opinion or historical records of first observation and not on population genetic surveys. However, new records are continuously published in the literature[42,47–50], also contributing to our knowledge on the spread of non-native ants[37,51], and compiled in the Global Ant Biodiversity Informatics (GABI)[32] to provide the latest developments on ant biogeography. Second, there are still many regions of the world are under-sampled[50,52]. This may affect the calculation of the $\beta$ diversity index, which partially depends on species richness. As a consequence, the degree of homogenization may be over-estimated in areas with much undiscovered ant biodiversity[42] because rare native species are more likely to be under-sampled than non-native species. The addition of new records as well as newly described species to particular regions would lower our estimated degree of homogenization. However, this is unlikely to affect our main conclusion that homogenization is heterogeneous and most pronounced on islands, as tropical and mainland regions mostly act as donor regions and islands mostly act as recipient regions of non-native ant species. However, future studies on the impact of ant invasions may analyze biogeographic patterns at finer resolution[53] to detect more precisely biogeographic transitions, as for recent studies on bioregionalization in European ants[54] and global native ant biodiversity[42]. Additionally, in our study we considered islands as entities that are smaller than a continent and surrounded by water (comprising both single islands and island archipelagos). We acknowledge that islands are largely under-explored with, for example, more than 108 large islands globally (with an area >200 km²) that have received no sampling effort[50]. This under-sampling may have affected our estimate of homogenization on islands, although we believe that the general pattern of homogenization along the tropical belt is likely to be robust. The release of a new database of global ant biodiversity on islands[50] is an exciting perspective for future research to investigate differences among different islands, linking the degree of homogenization to the characteristics of the islands (e.g., size, isolation, sovereign state) for example. More detailed species distribution data may also enable future studies to analyze the relative importance of potential drivers of changes in biogeographic patterns, such as climate or trade patterns.

Moreover, future work could assess changes in phylogenetic[54] and functional $\beta$ diversity once such data becomes available, giving complementary results about the potential evolutionary and ecosystem consequences of non-native species introductions. Finally, ant assemblages might be homogenized due to local extinction of endemic native species in addition to the establishment of widespread non-native species[15,17,24]. In our study, we did not assess the effect of extinctions as data on ant population declines are largely lacking (but see ref. 55), although their role is extremely intriguing.

In conclusion, ongoing globalization contributes to the spread of non-native species, with particularly important consequences for island assemblages. Species introductions are predicted to accelerate in the coming decades[6,56]. Therefore, global biodiversity

homogenization is likely to occur with unknown evolutionary, ecological, and economic consequences. As non-native species are among the greatest drivers of biodiversity loss globally[23,57–59], understanding the spatial variation and intensity of biotic homogenization more precisely is key to informing conservation measures[60] to preserve the biotic uniqueness of regions globally.

## Methods

### Distributional data and pre-processing

Species native ranges were sourced from the webmaps displayed on antmaps.org which is linked to the Global Ant Biodiversity Informatics (GABI) project[31,32]; the details of ant species distribution records are fully described in ref. 32. For non-native species, we used the native and non-native ranges of 309 non-native ant species that have been established outdoors, excluding non-native species that are only introduced indoors or intercepted at border controls, described in ref. 37. Our study did not require ethical approval.

From this dataset, we excluded species with unknown distributions as well as records which are listed as "dubious" or "needing verification". Species distributions were formatted as presence/absence data at the geographical scale of the sub-country political regions (referred to as "bentities" in GABI, hereafter polygons), and absences were inferred as the lack of presence data. The polygons, described in ref. 32, reflect human political delineations (e.g., country level; state), geology (e.g., mainlands, islands), and scientific knowledge (e.g., specific split of political entities). Out of a total of 546 polygons, ant species are found in 536.

Our final dataset comprised the distribution of 13,774 ant species with valid species name based on AntCat.org and additional non-native ranges of 309 non-native ant species. The native records of non-native ant species were considered to correspond to the species' ranges before human-mediated dispersal, while entire distribution including native and non-native ranges correspond to the species' current ranges after human-mediated dispersal.

We analyzed the distributions of 309 non-native ant species, including information on their native ranges (i.e., before human-mediated dispersal, 484 polygons, Supplementary Fig. 1a) and current ranges (i.e., after human-mediated dispersal, 512 polygons, Supplementary Fig. 1b). We additionally calculated the ratio of non-native ant species to total ant species richness by polygon (Supplementary Fig. 2).

### Geographical focus: global, mainlands, and islands

Among the 536 polygons where ant species are recorded, 384 were located on mainlands and 152 on islands. For this analysis, we defined an island as an area surrounded by water smaller than the smallest continent (with Greenland being therefore the biggest islands). To classify polygons as mainlands and islands, we used recent works on ant species distributions[50] and on invasive species[61] on islands (Supplementary Fig. 4). We did not considered Newfoundland as an island as most of its surface was comprised on mainlands. The number of native ant species varied from 0 to 962 on mainlands (mean ± sd = 175.8 ± 162.1) and 0 to 852 on islands (mean ± sd = 78.6 ± 146.2). The number of non-native ant species varied from 0 to 72 on mainlands (mean ± sd = 6.3 ± 7.7) and 0 to 59 on islands (mean ± sd = 14.5 ± 14.2).

### Identification of biogeographic realms based on compositional dissimilarity

We calculated pairwise matrices of compositional dissimilarity among polygons using the $\beta$ diversity index ($\beta_{sim}$, vegan package[62], v2.5-7). This metric is particularly suited for biogeographic studies because it measures species turnover by focusing on compositional differences more than differences in species richness ("narrow sense" turnover)[63]. The $\beta_{sim}$ index measures species turnover between two spatial entities based on presence/absence data, and ranges from 0 – total similarity – to 1 – total dissimilarity (1).

$$\beta_{sim} = 1 - \frac{a}{\min(b,c) + a} \tag{1}$$

Where a is the number of shared species between two geographic units, and b and c are the number of unique species in each of the two geographic units respectively.

To identify biogeographic patterns, we performed a clustering analysis of the compositional dissimilarity matrices using an unweighted pair group method with arithmetic mean (UPGMA) (hclust[64], stats package[65], v4.2.2)[13,53]. We applied the method to two distinct datasets: non-native ant species (309 species) and all ant species (13,774 species), both decomposed into species distributions before (native ranges) and after human-mediated dispersal (native + non-native ranges) of non-native species. To determine the number of clusters, we tested the stability of trees using a simple permutation test run on 999 iterations (rtest.hclust function based on ref. 66). Significant clustering was indicated by a $p$ value of an inferior node of less than 0.05. The reason for using this method is that it can identify the dissimilarity level below which all clusters can be considered non-random. To assess if our results are robust and do not change with the choice of the clustering methods, we also explored different frequently used clustering methodologies which revealed the same biogeographic patterns (elbow method[10], average silhouette[67], and Kelly–Gardner–Sutcliffe penalty[68]). We replicated the approach at the global level and for mainlands and islands separately.

The maps of non-native ant species compositional dissimilarity are displayed at the global level before and after the human-mediated dispersal of non-native species (Fig. 2 and Supplementary Fig. 5 for mainlands and islands). The maps of compositional dissimilarities of all ant species are displayed before and after the human-mediated dispersal of non-native species at all geographical foci (global: Fig. 2, mainlands and islands: Fig. 3).

To explore to what extent the size of species pools per polygon affects the delineation of biogeographic realms, we performed a sensitivity analysis. We performed separate cluster analyses to identify realm based on random selections of ant species (300, 400, 500, 1000, 2000, 5000, and 10,000 species among all ant species) to determine the minimum species pool size necessary to detect historical biogeographical pattern. Additionally, we tested if these realms can be detected using the native ranges of non-native ants (Supplementary Methods and Supplementary Fig. 3). This analysis revealed that their native ranges are representative of ant biogeography, as they correspond to the historical biogeographic realms for all ant species. This would not be the case for a random selection of 300 ant species, for which the biogeographic pattern would be much more variable. This is likely because non-native species have larger native ranges than other ant species (Supplementary Fig. 3).

### Patterns of homogenization/differentiation after human-mediated dispersal

To calculate the extent that a pair of polygons has been homogenized or has differentiated due to human-mediated dispersal of non-native ant species, we calculated a homogenization index[19] ($h_{i,j}$), where $h_{i,j} = \beta_{AHMD\,i,j} - \beta_{BHMD\,i,j}$, with $\beta_{AHMD\,i,j}$ representing the $\beta_{sim}$ index after human-mediated dispersal and $\beta_{BHMD\,i,j}$ the $\beta_{sim}$ index before human-mediated dispersal between polygons i and j. This index was calculated for all ant species at three geographical foci (global, mainlands and islands), for a total of three homogenization matrices. For each pairwise comparison, if $h_{i,j} > 0$ ($\beta_{AHMD\,i,j} > \beta_{BHMD\,i,j}$), the pair of polygons are subject to biotic differentiation (as the $\beta_{sim}$ index calculates how dissimilar two entities are) and if $h_{i,j} < 0$, there is biotic homogenization. We then calculated the proportion of assemblages that have been homogenized ($h_{i,j} < 0$) or differentiated ($h_{i,j} > 0$).

To assess which polygons are more prone to biotic homogenization, we calculated the average value of the homogenization index ($\bar{h}_{i\bullet}$,(2)) for each polygon across all pairwise comparisons (Fig. 4a). To test if $\bar{h}_{i\bullet}$ was linked to their location on islands and/or within the tropics, we used the Scheirer–Ray–Hare test (rcompanion package[69], v4.2.26) which is the equivalent of a non-parametric two-way ANOVA. A tropical versus non-tropical status was attributed to each polygon according to the location of each polygon centroid (sf package[70,71], v1.0-10). Polygons for which the centroid was located between the two latitudinal parallels 23° far from the equator were considered as tropical. We then mapped the average global homogenization $\bar{h}_{i\bullet}$ for all polygons (Fig. 4b).

$$\bar{h}_{i\cdot} = \sum_{j=1}^{N} h_{i,j}/N \qquad (2)$$

where N is the number of polygons (global: 536, islands: 152, mainlands: 384).

We tested if the number of non-native species exported from donor regions (i.e., the species' native ranges), and the number of non-native species imported by recipient regions (i.e., the species' non-native range) was linked to status as islands or mainlands and location within tropical or non-tropical areas, using a Wilcoxon signed-rank test with Bonferroni correction (Fig. 4c). To account for geographic non-independence of polygons, we then used separate GLMMs for donor and recipient regions in which we included "region" (i.e., 23 subcontinental regions as classified in the GABI database) as a random-effect term. We fitted the GLMMs using the Automatic Differentiation Model Builder GLMMADMB R package[72] (v0.8.3.3) which provides a framework to model over-dispersed data and zero inflation[27]. For each of the GLMMs, we tested both a Poisson and a negative binomial distribution, and in all cases, the latter produced a better fit based on AIC. The best model for the number of imported species per recipient region did not include the interactions between locations on islands and/or within the tropics. However, the best model for the number of exported species per donor region included the interaction ($p < 0.05$).

### Distance-decay relationship before and after human-mediated dispersal of non-native species

Areas that are geographically closer tended to have more similar species assemblages. We tested if the distance-decay relationship changed after human-mediated transport using non-linear least squares models of compositional similarity ($1-\beta_{sim}$) as a function of distance between polygon centroids (nls[73], stats package[65], v4.2.2) for all ant species (Fig. 5) at the global scale. We then used the Mantel statistic to test if the relationship between species assemblage similarity and geographic distance changed after human-mediated dispersal (mantel test, vegan package[62], v2.5-7).

### Reporting summary

Further information on research design is available in the Nature Portfolio Reporting Summary linked to this article.

### Data availability

The raw data that support this study were sourced from the webmaps displayed on antmaps.org which is linked to the Global Ant Biodiversity Informatics (GABI) database[31,32] and ref. 37. All processed data generated and analyzed in this study have been deposited in a Figshare repository accessible at https://doi.org/10.6084/m9.figshare.22188208.v1[74]. Source data are provided with this paper.

### Code availability

The full reproducible code is available at https://doi.org/10.6084/m9.figshare.22188208.v1[74]. Data processing and statistical analyses were undertaken in R (v.4.1.0; R Core Team, 2021) and RStudio (Version 2022.12.0 + 353). Graphics and maps were produced using the ggplot2[75] (v.3.4.1) and sf[70,71] (v1.0-10) packages.

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

## Acknowledgements

This research was funded by the Swiss National Science Foundation: grant 310030_192619 (CB) and TMPFP3_209715 (LAG), and the canton Vaud and the Fondation Sandoz-Monique de Meuron pour la relève universitaire (CB). Global, mainland and island icons used in Figs. 1, 2, 3 and Supplementary Figs. 3, 5, 6 has been designed using assets from Freepik.com.

## Author contributions

L.A.G., S.O., and C.B. designed the research. L.A.G. performed the research. S.O. and L.A.G. compiled the data and L.A.G. analyzed the data. L.A.G., S.O., and C.B. all contributed to the writing of the paper.

## Competing interests

The authors declare no competing interests.
