## [Peer Review File · Nature Communications]

Non-native ants are breaking down biogeographic boundaries and homogenizing community assemblagesEditorial Note: Parts of this Peer Review File have been redacted as indicated to remove third-party material where no permission to publish could be obtained.

REVIEWER COMMENTS

Reviewer #1 (Remarks to the Author):

The authors investigate the impact of alien species (those transported by human activity outside their native ranges, where they become a permanent addition to local communities) on the biogeographic boundaries of global biodiversity. They base this analysis on comprehensive data on ant species distributions with which they examine the biogeographic groupings of regions worldwide for alien-only and native and alien taxa combined, before and after human dispersal. The authors also examine changes in biotic homogenization (decreases in compositional dissimilarity levels) or differentiation (increases) for all regions, as well as for island and mainland regions in isolation. Additionally, they examine if the quantitative changes observed were driven by alien species being imported to the regions or by being exported from them. Finally, they measure the effects on distance decay patterns of compositional similarity.

The study reveals several global-scale patterns of strong interest to biogeographical and biodiversity change research on the effects of the Anthropocene. Most significantly, the authors find that human-mediated dispersal of some species has greatly homogenized the composition of species assemblages worldwide, leading to the biogeographical convergence of regions, particularly along the tropical belt. This finding is exciting, as it demonstrates how human activity during just a few centuries is reshaping natural patterns that have lasted for many millions of years. The effects are observed not only for the subset of alien species but for the whole species assemblages, i.e., regardless of their nativity status.

I believe this work has significant implications for the field and should be of interest to a broad audience (i.e., beyond the research field and related fields). Moreover, it is very clearly written, uses standard analytical procedures in the field, and the range of analyses is quite comprehensive (but see general comments below). Finally, all elements necessary to

replicate the analysis appear to be provided or publicly available (code and data).

In sum, I have a very positive opinion about this work and its importance to the field and beyond.

General comments:

The authors used a hierarchical clustering method (UPGMA), where the definition of clusters depends on the definition of a dissimilarity threshold. The authors mentioned a permutation test to determine the number of clusters to consider, but a deeper explanation and discussion of the threshold values obtained would improve the manuscript. A sensitivity analysis on the effects of changing the threshold values could offer insight into the robustness of the results (i.e., the emerging biogeographical boundaries).

Second, it would be interesting to have some analysis or at least some discussion on the drivers of the observed changes. For instance, what are the factors leading to the emergence of the pan-tropical region observed? Could climatic filtering acting upon the climatic tolerances of introduced species be a (dominant) contributing factor? Or is this a likely reflection of global trade patterns? Even if a detailed analysis of these factors falls beyond the scope of the work, their discussion would be relevant.

Minor comments:

The two 'with's' in the first sentence of the abstract makes it hard to read (I had to read it twice before reaching the intended meaning). Consider rephrasing.

Line 24: "As a consequence, alien species might break down historical biogeographic boundaries^{10,11}, separating distinct species assemblages (bioregions)."

This sentence could suggest that alien species may separate distinct species assemblages, while I believe this is being mentioned in regard to 'biogeographic boundaries'.

Line 64: "In this study, we have quantified both the changes to biogeographic barriers..."

Do you mean 'biogeographic boundaries'?

Line 66: "We separately analyzed biogeographic patterns before and after human mediated

dispersal for alien species assemblages (292 species) and for all ant species assemblages (13,758 species).” Suggest changing to: “We separately analyzed biogeographic patterns before and after human mediated dispersal for alien species assemblages (292 species) and for assemblages of all species (13,758 species).”

Line 255: ‘compositonnal’.

Line 287: For each of the GLMMs, we tested both a Poisson and a negative binomial distribution, and in all cases, the latter produced a better fit.

Better fit based on what? AIC? Please clarify.

Reviewer #2 (Remarks to the Author):

This paper compares global biogeographic regionalizations for ants estimated with known alien species and all species together. The authors found that alien species are responsible for increasing homogenization around the globe, particularly for the tropics and islands. The topic of biotic homogenization is important given increasing global change, especially for ants, which have strong links to ecosystem functions.

Although I think this paper does need numerous revisions, which I detail below, aside from one major point (see the below line-by-line comments with asterisks), I did not find major problems with the analyses or results described. What I think need revising/rethinking are 1) some general statements about the background and results, 2) descriptions of the methods, 3) presentation of the figures.

Other major points:

- 1) Please state what your hypotheses were at the end of the Introduction. What did you expect to find and why?
- 2) Throughout, please cite all R packages mentioned in the text. They are perhaps more important for your study than some of the papers cited, so they deserve recognition.

Line-by-line comments:

L24: Why would changes to beta diversity resulting from invasions of alien species create barriers to dispersal? And what exactly are these barriers in relation to? Please specify.

L25: If biogeographic boundaries are broken down, how would assemblages get "separated"? Wouldn't they be merged, in that assemblages would become more diffuse?

L28: See below, but perhaps you should focus on "to what extent" and not "if this happens", because "if" seems straightforward.

L32-34: I think that this question is fairly obvious -- more invasions of alien species will almost certainly lead to homogenization of assemblages. Perhaps the more relevant question without a clear answer is the "to what extent" part. How homogenized do assemblages get? How does this change with the focal taxon and geography considered?

L35: This seems to be what was just discussed in the previous paragraph: changing species assemblages affecting biogeographic boundaries. You had defined bioregions as distinct species assemblages in L25-26, so the previous paragraph seems to be about changing assemblages leading to changing biogeographic boundaries. Thus, I am confused by this intro sentence seemingly presenting a new concept. Perhaps better differentiate the concept of biogeographic boundaries from species assemblages? Aren't these boundaries products of evolution, geology, etc., and not just a delineation of where assemblages differ? As community assemblages are kind of a nested concept in biogeographic boundaries, I think the explanation here about assemblages should probably come before talk about bioregions to make the flow of concepts more linear.

L39: So does "biotic differentiation" restore biogeographic boundaries? I know that is likely doesn't, but you should explain why not. Also, is this an established term that means "differentiation of assemblages due to invasions of different invasive species"? Because if it doesn't, I'd explain this more clearly and avoid invoking a term like this (which may be more general than intended). But perhaps most importantly, this sentence breaks the flow of your explanation of biotic homogenization, so I'd move it.

L41: "and" island birds

L41-42: Remove this sentence? It's a bit repetitive.

L44: assessing consequences for "native species as well"

L45: What previous work? Please include refs.

L48: Differ how? The impact should be "stronger" on islands. Please make this clear.

L49-50: The fact that islands are more vulnerable is pretty established. Perhaps: However, as islands are in general more vulnerable to human impacts, they tend to be studied separately in the field of biogeography.

L50: It is not unknown if different parts of the world are being homogenized -- this is documented as you said earlier. Please rephrase.

L58: with "at least" 292 species

L60: Are these costs for the globe as a whole? Please specify.

L67: Perhaps: dispersal for assemblages of alien ant species (n = 292) and all described ant species (n = 13,758).

L68-69: It would make sense to mention the new Kass et al. paper here and say that distributions for all these species were also modeled at the grid-cell level.

L80: Very important to mention here that this is before human-mediated dispersal.

L87: This is an incomplete sentence.

L88-89: The Palearctic east-west separation was there before you added the invasive points.

**L93-94: This is an important point -- don't species with high sample size disproportionately influence the delineations? If species with known invasions have many more occurrence data than most other species (likely true), doesn't this bias the delineations?

**L96: Again, this would not be surprising if the point above is true. If so, the only way to tell if there is still a significant effect of these few invasive species is to do a randomization test where you randomly select balanced sets of occurrence data and compare them.

L116: "similar" to the pattern...

L119: is "that" the ratio...

L133: Please say here that you used a "non-parametric two-way ANOVA (see Methods)", which will be more familiar to most readers, then describe the kind of test you did in the Methods.

L138: When you say "assemblages" here, you mean those among the mainland and islands, or "across all assemblages homogenization increased". Thus, you cannot say that the assemblages "exported" or "imported" species, because now you are talking about the particular assemblages of the regions themselves. Please reword so this makes sense.

L143: Here too, same problem as above. The mainland did not "homogenize" because it

exported species. The globe became more homogeneous as a result. Only when you compare one thing to another (i.e., the mainland to the islands) can you invoke homogenization.

L146: Important to mention that most of these are endemics.

L149: I don't see clear evidence for this in Fig S2. Where is the evidence that the mainlands did not receive many invasive species? Is this relative to how many islands received?

L152-153: Thus far you have not focused on individual species or their idiosyncrasies at all, so this is a confusing wrap-up sentence. And what does it mean to say that "the process depends on... both exports and imports of species"? Can you please explain in more detail what point you are trying to make here?

L162-167: This belongs in the Discussion.

L170: millions of years of "natural dispersal" and evolution (?)

L180: Something more to the point would be to state that they include "add percentage" of the world's biodiversity hotspots.

L181: Why would smaller islands have more adaptive radiation? Wouldn't larger islands produce more endemic species from a single dispersal event?

L185: Kass et al. would be a more up-to-date ref for this.

L211: The GABI database includes only presence data. You are inferring absence as the lack of presence data. Please make this assumption clear.

L218-220: How did you attribute occurrence points as "native range" and "invasive range"? Please specify.

L222: You never explain where the bioregions data come from.

L223-224: Not clear what this means: "a non-random subset of all ant species". In relation to what?

L226: Why would a random selection of ant species not correspond to historical bioregions for all ants? This is confusing. Do you mean that you delineated bioregions based on the subset of invasive ants and then random selections, and determined that the invasive ants better recovered historical biogeographical patterns? If so, please explain this here, as it is not clear (the refs to Supp 3 are not sufficient to help the reader understand what was done).

L255: spelling

L258: plural of focus is "foci"

L275: each "polygon"

L276: for which "the" centroid

L285: Why GLMMs? Did you specify random effects? If so, what were they?

L296: Please spell out "non-linear least squares".

L298: Change to "nlm, package stats"?

L304: RStudio is not typically cited because the analysis does not rely on it.

Fig 1: What do the solid and dashed lines mean in the first step? The "current range" matrix is confusing because it looks like the alien range is very close to the native range, which is typically not the case. Can you separate them more?

The "Geographical focus section is repetitive. It seems there must be a more straightforward way to organize these first two sections than separating "alien" and "all species" into two separate columns, as the workflow is not different for each. Third section should be "Bioregion delineations". The term "polygons" by itself is not useful here, so perhaps instead use "regional polygons" or something. What is "K"? What is that double arrow symbol, and why is the midsection of the clustering plot designated? What do the colors mean in the bottom plots, and what are they showing? This figure needs descriptive text (brief) for each step, else a lengthy description in the caption. The figure must be self-explanatory, and currently it is not. Also fix spelling (compositional).

Fig 2: Please try to use similar colors for the top and bottom. For example, light green is only in the top plots, and seems to be replaced with pink in the bottom. It would be easier to compare the top and bottom if light green (or pink) represented Asia and the southern Pacific in both.

Fig 4: Is c) really a spatial representation? It is a plot with two axes that are not geographic coordinates, so I would not call it "spatial". Also, in the text there seems to be a statistical test on this data (GLMM?) that points to this plot (L142). The predictions of this model (lines) should be shown on the figure because it's hard to tell from the points alone what the relationships are.

Supplemental code: Looks like you include some LaTeX code in your R script to designate math syntax, but it is just confusing here unless the reader is familiar with this syntax. Please rewrite in a simpler notation.

Reviewer #3 (Remarks to the Author):

Main comments

In the manuscript entitled “Alien ants break down biogeographic boundaries and homogenize community assemblages in the Anthropocene”, Aulus-Giacosa and collaborators studied the effects of biological invasions on main biogeographic realm classification in ants. After testing the overall pattern, the authors tested it more specifically for insular and mainland regions as well as for temperate and tropical regions, while attempting to quantify the level of homogenization for these groups on the basis of importation and exportation of alien ant species.

The topic presented here is already well-known from several studies, mentioned by the authors themselves, such as Capinha et al. 2015 or Liu et al. 2021; but also from others not mentioned on plants using both taxonomic and phylogenetic approaches (Yang et al. 2021), amphibians, birds & mammals (Bernardo-Madrid et al. 2019) or on ecological interactions (Fricke & Svenning 2020), so it sounds that the authors just offer to add one more taxon from a list already including a wide range of taxa. The disproportionate invasion rate on islands in ants, as for other taxa, is also well-known (e.g. Moser et al. 2018), so it is not exactly surprising to perceive a stronger effect there. The novel aspect of this work is thus on the quantification of importer/exporter roles in the homogenization process for the main biogeographic realms. This aspect, however, would need to be considerably strengthened as this rely on particular definitions of the native and introduced ranges for the different 292 alien species considered, with several of them possessing uncertain native ranges at this point. It would thus require justification for each species, with the consideration of alternative scenario for problematic species.

The biogeographic results presented in the figure 2C are very surprising, with absence of distinctions between the Malagasy from the Afrotropical realm, Australian and Oceanian realms from the Oriental realms; but the presence of a Patagonian realm. This clearly differs from previous studies including relatively recent ones on vertebrates (see Holt et al. 2013).

The authors should note that Wallace's work while absolutely seminal and laudable, is nonetheless now outdated and more recent work should be referred to.

In my opinion, a major issue comes with your treatment of islands which represent very different units, not necessarily comparable with one another nor with mainland polygons. Indeed, what you refer as islands are in fact island groups in most cases, with some spread over hundreds if not thousands of kilometers (e.g. Fiji, Tuamotu Islands, Marquesas Islands, Lesser Antilles), contrary to mainland regions which form continuous units with their classification relying sometimes on geographic reasons, but for others on political ones. In addition, some islands may be composed of a single island, while others are constituted of hundreds of islands presenting very distinct dynamics of native and introduced species (and potential for future establishments). Additionally, the part on how you classified islands is unclear (Supplement lines 69-83). Why are some larger islands (Corsica, Jamaica, Java, Sardinia, Sicily, Sulawesi, Taiwan, Timor...) not considered as islands? Please explain further how the distinction between the islands and mainland groups were done. Based on my current understanding, it seems that ultimately, the comparison may be between large areas versus small areas rather than just islands vs mainland regions.

It should also be noted that islands are particularly under-sampled which is likely to strongly impact the pattern observed. In general, while you briefly discuss potential problems of under sampling in the discussion (lines 184-187), your analyses do not really cope with this issue. This is nonetheless critical on the way under-sampling could impact the overall robustness of bioregionalization at the global scale for both islands and mainland regions. You should note that in the past years, a number of studies on ants have pushed the frontier of biogeographic knowledge on this model organism by providing data at a much higher geographic resolution for both islands and mainland areas (Liu et al. 2023; Kass et al. 2022) while providing a greater understanding of the sampling gaps existing in these regions. Here the approach taken in your study does not include these progresses and limits the value of your results.

Additional information is also needed for the distinction between temperate vs. tropical regions.

I note after downloading the data from Wong et al. (2023) from which the alien ant data originated, and filtering for established species only, I was unable to retrieve the same data as those presented in your study. For instance, the maximum established species for

mainland and islands were of 72 (Florida) and 58 (Hawaii) respectively versus 68 and 61 presented in your study (Lines 235-236). Please explain this discrepancy and if you have done modifications to the data, how those were conducted.

Minor comments

All over your text, you would need to refer to biogeographic realms and not “regions” as those refer to a different unit.

Line 18: the dispersal of alien species is of both accidental and deliberate nature, as thousands species of plants, vertebrates and invertebrates have been spread for economic reasons (agriculture, forestry, biological control, landscaping...).

Line 62-63: this sentence is factually incorrect as species of the *Formica rufa* group were introduced in Central Italy (see work by Pavan in the 1950's & 60's) and later on in Eastern Canada in the 70's.

Line 93: a detail but as you did not use all valid ant species in your analyses, this number should be 1.8% (292/15850)

Lines 180-183: see also Moser et al. (2018) which includes ants as well as other taxa.

Lines 190-192: I don't understand the point of this sentence as the authors acknowledge the benefit of increased spatial resolution but did not use the latest and most comprehensive data available. See also Kass et al. 2022 presenting a much finer spatial resolution and identification of undersampled regions.

Lines 192-195: see Yang et al. 2021

Lines 195-197: See direct examples in Bernardo-Madrid et al. 2019.

Lines 199: replace “severe” by “deep” as you did not measure any effects directly and local or regional extinctions consequences on native species unknown.

References:

Bernardo-Madrid et al. 2019. *Ecology Letters* 22, 1297–1305

Capinha et al. 2015. *Science* 348, 1248–1251

Fricke & Svenning 2020. *Nature* 585, 74–78.

Holt et al. 2013. *Science* 339, 74.

Kass et al. 2022. *Science Advances* 8(31), eabp9908.

Liu et al. 2021. *Curr. Zool.* 67, 393–402.

Moser et al. 2018. *PNAS* 115, 9270–9275.

Yang et al. 2021. *Nature Communications* 12, 7290.

We would like to thank the reviewers for their constructive comments and important concerns they raised which helped improving the manuscript. We also added some discussion about the limitations of the data we used in the manuscript. Please find below our detailed point-by-point replies (**in red) to the reviewer's comments (in black).

Reviewer #1 (Remarks to the Author):

The authors investigate the impact of alien species (those transported by human activity outside their native ranges, where they become a permanent addition to local communities) on the biogeographic boundaries of global biodiversity. They base this analysis on comprehensive data on ant species distributions with which they examine the biogeographic groupings of regions worldwide for alien-only and native and alien taxa combined, before and after human dispersal. The authors also examine changes in biotic homogenization (decreases in compositional dissimilarity levels) or differentiation (increases) for all regions, as well as for island and mainland regions in isolation. Additionally, they examine if the quantitative changes observed were driven by alien species being imported to the regions or by being exported from them. Finally, they measure the effects on distance decay patterns of compositional similarity.

The study reveals several global-scale patterns of strong interest to biogeographical and biodiversity change research on the effects of the Anthropocene. Most significantly, the authors find that human-mediated dispersal of some species has greatly homogenized the composition of species assemblages worldwide, leading to the biogeographical convergence of regions, particularly along the tropical belt. This finding is exciting, as it demonstrates how human activity during just a few centuries is reshaping natural patterns that have lasted for many millions of years. The effects are observed not only for the subset of alien species but for the whole species assemblages, i.e., regardless of their nativity status. I believe this work has significant implications for the field and should be of interest to a broad audience (i.e., beyond the research field and related fields). Moreover, it is very clearly written, uses standard analytical procedures in the field, and the range of analyses is quite comprehensive (but see general comments below). Finally, all elements necessary to replicate the analysis appear to be provided or publicly available (code and data). In sum, I have a very positive opinion about this work and its importance to the field and beyond.

****We would like to thank reviewer 1 for their positive opinion on the manuscript. Thanks to the detailed comments, we improved the manuscript notably, especially the description of the methods and the discussion.**

General comments:

The authors used a hierarchical clustering method (UPGMA), where the definition of clusters depends on the definition of a dissimilarity threshold. The authors mentioned a permutation test to determine the number of clusters to consider, but a deeper explanation and discussion of the threshold values obtained would improve the manuscript. A sensitivity analysis on the effects of changing the threshold values could offer insight into the robustness of the results (i.e., the emerging biogeographical boundaries).

** We thank the reviewer for pointing that out, and indeed raises the important question: how to select the appropriate number of clusters. We have now clarified the method in the manuscript (l.273-286). We agree that a sensitivity analysis on effects of different thresholds could give more confidence into the robustness of the results. Indeed, there are several methods available to detect clusters after building trees, and these methods may identify different numbers of clusters in a dataset. In our manuscript, we built the trees using a hierarchical clustering method (UPGMA) and identified clusters using the method of Greenacre (2011), for several reasons. First, it is based on the permutation of the initial tree (999. permutations) to statistically detect which node of the tree is stable. Second, after a preliminary analysis testing several other methods to determine the optimal number of clusters: elbow method based on variance (Holt et al. 2013), average silhouette width (ASW) with the R package cluster (Maechler et al., 2022), and Kelly-Gardner-Sutcliffe penalty (KGS) with the R package mptree (White & Gramacy, 2022), the different methods gave very similar results. We have now added both reasons in the manuscript to justify our method. Although the optimal number of clusters varied according to each method, the delineated biogeographic realms were very similar (see figure below) and the slight differences in number were mostly due to some polygons that were not changing the main pattern. Importantly, the method of cluster selection did not affect our conclusion on changing patterns before and after human-mediated dispersal of alien species.

The figure shows the delineated realms of all ant species (13,758 species), based on the four compared methods: Greenacre (Kg), elbow method or node height (Knh), average silhouette width (Kasw), and Kelly-Gardner-Sutcliffe penalty (Kkgs). The average silhouette width (Kasw) distinguishes more realms here (12 in total), four of them corresponding to a subdivision of the Neotropics and one corresponding to the subdivision of the Palearctic with a differentiation of the sub-Saharan. The Kelly-Gardner-Sutcliffe penalty method in this case delineates the lowest number of biogeographic realms, with India merged with the Oriental realm and the most southern part of South America merged into the Neotropics.

Second, it would be interesting to have some analysis or at least some discussion on the drivers of the observed changes. For instance, what are the factors leading to the emergence of the pan-tropical region observed? Could climatic filtering acting upon the climatic

tolerances of introduced species be a (dominant) contributing factor? Or is this a likely reflection of global trade patterns? Even if a detailed analysis of these factors falls beyond the scope of the work, their discussion would be relevant.

****We agree with the reviewer that the drivers of the observed changes are of interest although this is beyond the scope of the work. As suggested by the reviewer, we added some discussion on that point “We did not quantify the role of different environmental or socio-economic drivers of the observed changes, but as alien ant species mostly originate from and are introduced in tropical areas (GLMM, $p < 0.0001$, Fig.S2), climatic filtering is likely a main contributing factor in alien species establishment¹⁴. Additionally, trade – and in particular the plant and fruit trade - is known to be an important introduction pathway of alien ants⁴³ and could determine which locations within a suitable climatic area (the tropics) are more likely to be reached by alien ants^{13,14}.”**

Minor comments:

**** All the minor comments were considered as suggested for the reviewer and we thank the reviewer for pointing the mistakes out.**

The two ‘withs’ in the first sentence of the abstract makes it hard to read (I had to read it twice before reaching the intended meaning). Consider rephrasing.

****We rephrased.**

Line 24: "As a consequence, alien species might break down historical biogeographic boundaries^{10,11}, separating distinct species assemblages (bioregions)." This sentence could suggest that alien species may separate distinct species assemblages, while I believe this is being mentioned in regard to ‘biogeographic boundaries’.

****The sentence was rephrased for clarity.**

Line 64: "In this study, we have quantified both the changes to biogeographic barriers..." Do you mean ‘biogeographic boundaries’?

****Yes, this has been changed.**

Line 66: “We separately analyzed biogeographic patterns before and after human mediated dispersal for alien species assemblages (292 species) and for all ant species assemblages (13,758 species).” Suggest changing to: “We separately analyzed biogeographic patterns before and after human mediated dispersal for alien species assemblages (292 species) and for assemblages of all species (13,758 species).”

****We followed the reviewer’s suggestion and modified this sentence.**

Line 255: ‘compositonnal’.

**** We modified.**

Line 287: For each of the GLMMs, we tested both a Poisson and a negative binomial distribution, and in all cases, the latter produced a better fit. Better fit based on what? AIC? Please clarify.

****The fits of models were assessed by AIC. We thank the reviewer for pointing that out and we clarified.**

Reviewer #2 (Remarks to the Author):

This paper compares global biogeographic regionalizations for ants estimated with known alien species and all species together. The authors found that alien species are responsible for increasing homogenization around the globe, particularly for the tropics and islands. The topic of biotic homogenization is important given increasing global change, especially for ants, which have strong links to ecosystem functions.

Although I think this paper does need numerous revisions, which I detail below, aside from one major point (see the below line-by-line comments with asterisks (*)), I did not find major problems with the analyses or results described. What I think need revising/rethinking are 1) some general statements about the background and results, 2) descriptions of the methods, 3) presentation of the figures.

**** We thank the reviewer 2 for the very detailed and developed line-by-line comments made on the manuscript, which helped us improved the general flow of concepts (backgrounds, methods, and figures). We treated all the comments with attention and modified our manuscript accordingly. We also replied to the major concern of the reviewer marked with an asterisk: we agree that the reviewers concern was warranted and would have been indeed a problem if we were using occurrence data. But this was not the case. Therefore, we revised the explanations and provided more details in the methods concerning the type of data we used.**

Other major points:

1) Please state what your hypotheses were at the end of the Introduction. What did you expect to find and why?

**** We thank the reviewer for the comment about the clarity of the introduction and added the hypotheses in the end of the introduction accordingly:**

“Here, our aim is to test to what extent alien ant species dispersal changes biogeographical boundaries. Specifically, we test the hypothesis that a general trend toward biotic homogenization is accompanied by large regional differences, with stronger homogenization of on islands (due to their depauperate faunal composition and greater vulnerability to invasion²³) and tropical areas (since they are climatically similar to the native ranges of most alien ant species).” (l.75-80).

2) Throughout, please cite all R packages mentioned in the text. They are perhaps more important for your study than some of the papers cited, so they deserve recognition.

**** We agree with the pertinent comment made by the reviewer and recognize that authors of packages deserve recognition. Therefore, we added references related to the R packages.**

Line-by-line comments:

L24: Why would changes to beta diversity resulting from invasions of alien species create barriers to dispersal? And what exactly are these barriers in relation to? Please specify.

** We agree that the sentence was somehow misleading. We rephrased it, to mean that barriers to dispersal were historical and are expected to change according to alien species dispersal.

L25: If biogeographic boundaries are broken down, how would assemblages get "separated"? Wouldn't they be merged, in that assemblages would become more diffuse?

**We agree that the sentence was confusing and rephrased it: "Historically, the spatial turnover patterns in species assemblages (β diversity) were characterized by several abrupt transitions, called "biogeographic boundaries". One famous example of a biogeographic boundary is the Wallace line separating the Indomalayan and the Australasian realms. Biogeographic boundaries have been shaped by geography, past and present environmental differences and evolutionary history^{10,11}. However, the reshuffling of biodiversity with human-mediated transport has the potential to break these historical biogeographic boundaries^{11,12}."

L28: See below, but perhaps you should focus on "to what extent" and not "if this happens", because "if" seems straightforward.

** We thank the reviewer for this comment and changed the sentence accordingly: "However, it remains unclear to what degree this occurs in other taxonomic groups, [...]".

L32-34: I think that this question is fairly obvious -- more invasions of alien species will almost certainly lead to homogenization of assemblages. Perhaps the more relevant question without a clear answer is the "to what extent" part. How homogenized do assemblages get? How does this change with the focal taxon and geography considered?

** We agree with the reviewer's comment on the fact that the more relevant question of the manuscript is the "to what extent" regions are affected by alien species spread. Indeed, our aim was to quantify the level of homogenization for each assemblage (relative to all other assemblages) to test if particular regions (tropical areas, islands) are more strongly affected. We therefore modified the sentence (l.37, l.42) and have emphasized this point throughout the manuscript.

L35: This seems to be what was just discussed in the previous paragraph: changing species assemblages affecting biogeographic boundaries. You had defined bioregions as distinct species assemblages in L25-26, so the previous paragraph seems to be about changing assemblages leading to changing biogeographic boundaries. Thus, I am confused by this intro sentence seemingly presenting a new concept. Perhaps better differentiate the concept of biogeographic boundaries from species assemblages? Aren't these boundaries products of evolution, geology, etc., and not just a delineation of where assemblages differ? As community assemblages are kind of a nested concept in biogeographic boundaries, I think the

explanation here about assemblages should probably come before talk about bioregions to make the flow of concepts more linear.

**** We thank the reviewer and agree that assemblages should be better introduced before the idea of biogeographic realms. These borders between realms are indeed the products of evolution, geology, and other environmental factors. We have clarified this now (l.29-35).**

L39: So does "biotic differentiation" restore biogeographic boundaries? I know that is likely doesn't, but you should explain why not. Also, is this an established term that means "differentiation of assemblages due to invasions of different invasive species"? Because if it doesn't, I'd explain this more clearly and avoid invoking a term like this (which may be more general than intended). But perhaps most importantly, this sentence breaks the flow of your explanation of biotic homogenization, so I'd move it.

**** Thank you for this remark, we have now clarified the concepts of homogenization and differentiation. They are standard terminology in the study of species invasions where biotic homogenization refers to increasing similarity between areas and differentiation refers to the increasing dissimilarity between areas (see Olden et al. 2006, doi:10.1111/j.1365-2699.2006.01572.x, and Qian and Qian, 2022, DOI: 10.1111/ddi.13612).**

We modified the sentence to clarify this: "In parallel to these changes in biogeographic boundaries, the global movement of species may either lead to the homogenization or differentiation of species assemblages. Homogenization may happen if the same set of species is introduced in several regions which become increasingly similar in terms of species composition as a result. Alternatively, differentiation of assemblages¹⁵ could happen due to invasions of different alien species. "

L41: "and" island birds

L41-42: Remove this sentence? It's a bit repetitive.

L44: assessing consequences for "native species as well"

**** We changed accordingly these minor errors as suggested by the reviewer.**

L45: What previous work? Please include refs.

**** We agree with the reviewer that it could have been clearer that the reference was made according to the cited studies on biotic homogenization, regarding their specific limitations (l.47-53). We changed the sentence accordingly.**

L48: Differ how? The impact should be "stronger" on islands. Please make this clear.

**** This sentence has been changed.**

L49-50: The fact that islands are more vulnerable is pretty established. Perhaps: However, as islands are in general more vulnerable to human impacts, they tend to be studied separately in the field of biogeography.

**** We thank the reviewer for his suggestion and changed the sentence accordingly.**

L50: It is not unknown if different parts of the world are being homogenized -- this is documented as you said earlier. Please rephrase.

****We thank the reviewer for this comment. We acknowledge that homogenization of assemblages is already documented. However, the rate of homogenization by location is unknown. To clarify this point, we modified the sentence.**

L58: with "at least" 292 species

L60: Are these costs for the globe as a whole? Please specify.

L67: Perhaps: dispersal for assemblages of alien ant species (n = 292) and all described ant species (n = 13,758).

L68-69: It would make sense to mention the new Kass et al. paper here and say that distributions for all these species were also modeled at the grid-cell level.

L80: Very important to mention here that this is before human-mediated dispersal.

L87: This is an incomplete sentence.

L88-89: The Palearctic east-west separation was there before you added the invasive points.

**** We thank the reviewer for pointing out those seven minor errors and have corrected them now.**

(*)L93-94: This is an important point -- don't species with high sample size disproportionately influence the delineations? If species with known invasions have many more occurrence data than most other species (likely true), doesn't this bias the delineations?

(*)L96: Again, this would not be surprising if the point above is true. If so, the only way to tell if there is still a significant effect of these few invasive species is to do a randomization test where you randomly select balanced sets of occurrence data and compare them.

**** We agree that invasive species are more studied and may have more occurrences recorded than most other species. However, here we did not directly use occurrences, but recorded species' presence (at least one occurrence point) or absence (no occurrence points) in each polygon to build a presence/absence matrix. We have now clarified this in the text (l.237-239). However, we recognize that our knowledge of species distributions may vary according to the species identity (remark pointed out by reviewer 3) and we added a specific discussion on the limitations of the data we used, as suggested by the Editor (l.201-218).**

L116: "similar" to the pattern...

****We modified.**

L119: is "that" the ratio...

**** We modified.**

L133: Please say here that you used a "non-parametric two-way ANOVA (see Methods)", which will be more familiar to most readers, then describe the kind of test you did in the Methods.

** We thank the reviewer for this remark and changed the sentence accordingly.

L138: When you say "assemblages" here, you mean those among the mainland and islands, or "across all assemblages homogenization increased". Thus, you cannot say that the assemblages "exported" or "imported" species, because now you are talking about the particular assemblages of the regions themselves. Please reword so this makes sense.

** We thank the reviewer for pointing out that the meaning of assemblages in this sentence was unclear. We have modified that.

L143: Here too, same problem as above. The mainland did not "homogenize" because it exported species. The globe became more homogeneous as a result. Only when you compare one thing to another (i.e., the mainland to the islands) can you invoke homogenization.

** As the previous comment, we agree it was unclear and modified it accordingly (l. 157-159). "In contrast, at a global level the homogenization index of mainlands increases because mainlands are greater donors of alien species".

L146: Important to mention that most of these are endemics.

** After reclassifying Madagascar as island (following comments by reviewer 3), this part of the discussion was removed from the manuscript, although we agree that more than 80% of Madagascan species are endemics.

L149: I don't see clear evidence for this in Fig S2. Where is the evidence that the mainlands did not receive many invasive species? Is this relative to how many islands received?

** We thank the reviewer for this comment and realized that the supplementary figure cited was not accurate and we corrected this. Mainlands have received invasive species, but the fraction of alien on native remains relatively low (Fig. S3). We have clarified this now (l.160-163).

L152-153: Thus far you have not focused on individual species or their idiosyncrasies at all, so this is a confusing wrap-up sentence. And what does it mean to say that "the process depends on... both exports and imports of species"? Can you please explain in more detail what point you are trying to make here?

** We thank the reviewer for pointing that out and have reformulated the sentence. "Indeed, the global homogenization process depends on both exports (by donor regions) and imports (by recipient regions) of alien species."

L162-167: This belongs in the Discussion.

** We agree and moved this in the two first paragraphs of the general discussion.

L170: millions of years of "natural dispersal" and evolution (?)

**** We added the term as suggested by the reviewer.**

L180: Something more to the point would be to state that they include "add percentage" of the world's biodiversity hotspots.

**** We agree and modified the sentence accordingly "This is particularly concerning as almost two thirds of biodiversity hotspots³⁵ are located in tropical regions and islands are well-known centers of endemism^{41,42}."**

L181: Why would smaller islands have more adaptive radiation? Wouldn't larger islands produce more endemic species from a single dispersal event?

**** We thank the reviewer for the question and decided to delete the sentence.**

L185: Kass et al. would be a more up-to-date ref for this.

**** We agree.**

L211: The GABI database includes only presence data. You are inferring absence as the lack of presence data. Please make this assumption clear.

**** We acknowledge that we used presence-only data, and inferred absence as the lack of presence data. We modified the sentence as suggested (l.237-239).**

L218-220: How did you attribute occurrence points as "native range" and "invasive range"? Please specify.

**** We explained this in more detail now: "For each record, we extracted the species' range status as native or alien from the GABI database. The native records of alien species were considered to correspond to the species' ranges before human-mediated dispersal, while the sum of native and alien records corresponds to the alien species' current ranges after human-mediated dispersal." (l.244-248).**

L222: You never explain where the bioregions data come from.

**** We thank the reviewer for the comment which highlights that we were not clear enough about the method used to delineate biogeographic realms. We did not use "bioregion data" but delineated biogeographic realms based on the dissimilarity of global species assemblages. We decided to be more explicit about this and have moved a part explaining this more carefully to the method section and called it "Identification of biogeographic realms based on compositional dissimilarity".**

L223-224: Not clear what this means: "a non-random subset of all ant species". In relation to what?

** We agree and modified the sentence: “We performed separate cluster analyses to identify realm based on random selections of ant species (300, 400, 500, 1000, 2000, 5000, and 10,000 species among all ant species) to determine the minimum species pool size necessary to detect historical biogeographical pattern.”

L226: Why would a random selection of ant species not correspond to historical bioregions for all ants? This is confusing. Do you mean that you delineated bioregions based on the subset of invasive ants and then random selections, and determined that the invasive ants better recovered historical biogeographical patterns? If so, please explain this here, as it is not clear (the refs to Supp 3 are not sufficient to help the reader understand what was done).

** We agree with the reviewer that it was unclear. We wanted to test if the native ranges of alien ants are representative of all ant biodiversity, *i.e.*, if they recapitulate the same broad biogeographic realms. However, since alien ant species represent a relatively small number of species compared to all ants, we wondered if a small species pool could be sufficient to detect biogeographic boundaries. We expected that a minimum random species pool size was necessary to capture historical biogeographic realms of all ant species. We have added this information to the manuscript (l.292-302). “To explore to what extent the size of species pools per polygon affects the delineation of biogeographic realms, we performed a sensitivity analysis. We performed separate cluster analyses to identify realm based on random selections of ant species (300, 400, 500, 1000, 2000, 5000, and 10,000 species among all ant species) to determine the minimum species pool size necessary to detect historical biogeographical pattern. Additionally, we tested if these realms can be detected using the native ranges of alien ants (see Supplementary 3).”

L255: spelling

L258: plural of focus is "foci"

L275: each "polygon"

L276: for which "the" centroid

** We thank the reviewer and corrected this.

L285: Why GLMMs? Did you specify random effects? If so, what were they?

** We thank the reviewer for pointing out that we did not specify the random effects. Indeed, we used a GLMM to model over-dispersed data in which we included “region” (*i.e.*, 23 subcontinental regions as classified in the Global Ant Biodiversity (GABI) database) as a random-effect term to account for the geographic non-independence. We used separate GLMMs to explain variation in alien species richness in their native range (donor regions) and alien species richness in their alien range (recipient regions). We included climatic status (tropical vs. non-tropical), geographic status (mainlands vs. islands) as well as the interaction of those variables. We fitted all models with both Poisson and negative-binomial distributions, and selected the best model as assessed by lower AIC values. We changed this accordingly in the manuscript and clarified the method.

L296: Please spell out "non-linear least squares".

**** Done.**

L298: Change to "nls, package stats"?

**** Done.**

L304: RStudio is not typically cited because the analysis does not rely on it.

**** We agreed with the reviewer and decided to remove it.**

Fig 1: What do the solid and dashed lines mean in the first step? The "current range" matrix is confusing because it looks like the alien range is very close to the native range, which is typically not the case. Can you separate them more?

The "Geographical focus section is repetitive. It seems there must be a more straightforward way to organize these first two sections than separating "alien" and "all species" into two separate columns, as the workflow is not different for each. Third section should be "Bioregion delineations". The term "polygons" by itself is not useful here, so perhaps instead use "regional polygons" or something. What is "K"? What is that double arrow symbol, and why is the midsection of the clustering plot designated? What do the colors mean in the bottom plots, and what are they showing? This figure needs descriptive text (brief) for each step, else a lengthy description in the caption. The figure must be self-explanatory, and currently it is not. Also fix spelling (compositional).

**** We thank the reviewer for these suggestions. Concerning the first comment on native and alien range, we modified the "current range matrix" to illustrate the fact that alien range may be near or far from native range. We reorganized the first two sections into three, separating input data, data preparation and geographical focus to avoid repetition in the workflow. Spelling and grammar have been fixed. We decided to keep the term polygons as it is the term used throughout the manuscript, but according to the remark on descriptive text decided to explain further in the caption. We also modified the "realms delineations" section to make the method clearer in the figure (double arrow symbol for the permutation test). We removed K as it was not giving any important information (K = number of stable clusters = number of selected biogeographic groups). We also, as suggested, defined the meaning of the colors both on the final tree and map.**

Fig 2: Please try to use similar colors for the top and bottom. For example, light green is only in the top plots, and seems to be replaced with pink in the bottom. It would be easier to compare the top and bottom if light green (or pink) represented Asia and the southern Pacific in both.

**** We modified accordingly to the remark.**

Fig 4: Is c) really a spatial representation? It is a plot with two axes that are not geographic coordinates, so I would not call it "spatial". Also, in the text there seems to be a statistical test on this data (GLMM?) that points to this plot (L142). The predictions of this model (lines) should be shown on the figure because it's hard to tell from the points alone what the relationships are.

** We agree with the reviewer that the figure was not a spatial representation, and that it was not cited at the right place. However, because it was a bit hard to follow the flow of ideas, we decided to update the figure towards a simpler one (Fig. 4c), illustrating that donor regions are located on mainlands, with an effect of mainlands tropical status, whereas recipients' regions were tropical islands. Differences between groups were tested using a Wilcoxon test and results are illustrated on the new figure.

Supplemental code: Looks like you include some LaTeX code in your R script to designate math syntax, but it is just confusing here unless the reader is familiar with this syntax. Please rewrite in a simpler notation.

** We modified accordingly.

Reviewer #3 (Remarks to the Author):

Main comments

In the manuscript entitled “Alien ants break down biogeographic boundaries and homogenize community assemblages in the Anthropocene”, Aulus-Giacosa and collaborators studied the effects of biological invasions on main biogeographic realm classification in ants. After testing the overall pattern, the authors tested it more specifically for insular and mainland regions as well as for temperate and tropical regions, while attempting to quantify the level of homogenization for these groups on the basis of importation and exportation of alien ant species.

The topic presented here is already well-known from several studies, mentioned by the authors themselves, such as Capinha et al. 2015 or Liu et al. 2021; but also from others not mentioned on plants using both taxonomic and phylogenetic approaches (Yang et al. 2021), amphibians, birds & mammals (Bernardo-Madrid et al. 2019) or on ecological interactions (Fricke & Svenning 2020), so it sounds that the authors just offer to add one more taxon from a list already including a wide range of taxa. The disproportionate invasion rate on islands in ants, as for other taxa, is also well-known (e.g. Moser et al. 2018), so it is not exactly surprising to perceive a stronger effect there. The novel aspect of this work is thus on the quantification of importer/exporter roles in the homogenization process for the main biogeographic realms. This aspect, however, would need to be considerably strengthened as this rely on particular definitions of the native and introduced ranges for the different 292 alien species considered, with several of them possessing uncertain native ranges at this point. It would thus require justification for each species, with the consideration of alternative scenario for problematic species.

** We thank the reviewer for this comment. Indeed, accurate knowledge on native and alien ranges is a major issue to infer biogeographic realms. Few delimitations of native vs. alien ranges have been properly analyzed using population genetics and thus most range delimitations are based on expert opinions or historical data of first observation of alien species. We therefore agree that certainty on native ranges would require further investigation on a species-by-species base. But this is a huge undertaking requiring many years of ant sampling across the world and large-scale genetic analyses, which is unfortunately beyond our means. However, in this study we used the most up-to-date and large database on ant species distributions which is curated and regularly complemented by some of the most

knowledgeable experts on ant ecology and distributions (further information can be found in the Global Ant Biodiversity Database (GABI) published by Guenard et al. in 2017). We believe this dataset is currently the best species distribution database for any invertebrate taxon and is suitable to elucidate “to what extent alien species influence the biogeography of the entire taxon”. However, we agree with the fact that more detailed knowledge of ant species distributions would be of great interest for future research. We added a specific discussion on the limitations, also in accordance with the Editor’s comment about data limitations (l.201-218).

The biogeographic results presented in the figure 2C are very surprising, with absence of distinctions between the Malagasy from the Afrotropical realm, Australian and Oceanian realms from the Oriental realms; but the presence of a Patagonian realm. This clearly differs from previous studies including relatively recent ones on vertebrates (see Holt et al. 2013). The authors should note that Wallace’s work while absolutely seminal and laudable, is nonetheless now outdated and more recent work should be referred to.

** We thank the reviewer for this interesting comment. First, we acknowledge that there are more recent studies on the delineation of vertebrates’ realms. However, the scope of our paper was not to compare the delineation of realms between taxa, but rather compare the “before-after human-mediated dispersal of alien species” pattern in the delineation of biogeographic realms in ants. Knowing this, we used the known Wallacean realms as a base comparison for a reason of simplicity in the discussion of the results. Moreover, we were not fully convinced by the method used in Holt et al. 2013, as it was commented by Kreft and Jetz, 2013 (<https://doi.org/10.1126/science.1237471>). However, we cited the paper as a reference in the delineation of biogeographic realms.

Concerning the realms delineation, the figures have all been updated after we reclassified each polygon in the two categories: islands vs. mainlands.

In my opinion, a major issue comes with your treatment of islands which represent very different units, not necessarily comparable with one another nor with mainland polygons. Indeed, what you refer as islands are in fact island groups in most cases, with some spread over hundreds if not thousands of kilometers (e.g. Fiji, Tuamotu Islands, Marquesas Islands, Lesser Antilles), contrary to mainland regions which form continuous units with their classification relying sometimes on geographic reasons, but for others on political ones. In addition, some islands may be composed of a single island, while others are constituted of hundreds of islands presenting very distinct dynamics of native and introduced species (and potential for future establishments). Additionally, the part on how you classified islands is unclear (Supplement lines 69-83). Why are some larger islands (Corsica, Jamaica, Java, Sardinia, Sicily, Sulawesi, Taiwan, Timor...) not considered as islands? Please explain further how the distinction between the islands and mainland groups were done. Based on my current understanding, it seems that ultimately, the comparison may be between large areas versus small areas rather than just islands vs mainland regions.

** We entirely agree with the suggestion to reconsider the status of some polygons. To reclassify each polygon as “islands” or “mainlands” we used recent work on ant species distribution on islands (Liu et al., 2023) and the full islands database published as supplementary of Bodey et al. 2022. We defined an island as an area surrounded by water

smaller than the smallest continent (with Greenland being therefore the biggest islands). Islands in our dataset can comprise unique areas (single islands) or group of islands (archipelagos) based on the definition of polygons made in Guenard et al. 2017. The detailed information is given in the Methods (l.254-259).

We also agree with the reviewer that there are a large variety of polygons that are considered as islands, comprising archipelagos or single islands, based on the delineation of polygons available in the database we used. It is true that island delineation is sometimes based on geographic constraints, sometimes on political dependencies. However, the availability of the data constrained the analysis in that way. This would be concerning if our aim were to compare the impact of alien species among different islands, but our aim was to test to what extent alien ant species dispersal changes biogeographical boundaries, and to assess the trend of greater global homogenization on islands than on the mainland.

It should also be noted that islands are particularly under-sampled which is likely to strongly impact the pattern observed. In general, while you briefly discuss potential problems of under sampling in the discussion (lines 184-187), your analyses do not really cope with this issue. This is nonetheless critical on the way under-sampling could impact the overall robustness of bioregionalization at the global scale for both islands and mainland regions. You should note that in the past years, a number of studies on ants have pushed the frontier of biogeographic knowledge on this model organism by providing data at a much higher geographic resolution for both islands and mainland areas (Liu et al. 2023; Kass et al. 2022) while providing a greater understanding of the sampling gaps existing in these regions. Here the approach taken in your study does not include these progresses and limits the value of your results.

** We thank the reviewer for this comment. We agree that sampling bias changes ant species richness estimates, but we believe that under-sampled areas have less records of both native and alien species. Moreover, the goal of our study was not to model the potential distribution of species using species distribution modeling (such as in Kass et al. 2022), but rather using the raw distribution knowledge trying to understand to what extent human-mediated dispersal modifies biogeographic patterns and homogenizes ant assemblages. We believe that the general observed pattern at this scale would not drastically change with higher sampling on islands.

Additionally, the reviewer is correct that data with higher geographic resolution has recently been released for islands (Liu et al. 2023), presenting separately some single islands belonging to the same archipelago. Finer geographic division (of island archipelagos or mainland entities) may allow delineating biogeographic boundaries more precisely, but it will not change the global pattern of homogenization that we report in this study. Here, we describe broad patterns of regional differences in homogenization, comparing islands vs mainlands and tropical vs non-tropical regions. It is an exciting perspective for future research to investigate differences among different islands, linking the degree of homogenization to the characteristics of the islands for example. Indeed, this is beyond the scope of our study, but we have added this perspective to the discussion.

Additional information is also needed for the distinction between temperate vs. tropical regions.

** We thank the reviewer and clarified (l.318-321).

I note after downloading the data from Wong et al. (2023) from which the alien ant data originated, and filtering for established species only, I was unable to retrieve the same data as those presented in your study. For instance, the maximum established species for mainland and islands were of 72 (Florida) and 58 (Hawaii) respectively versus 68 and 61 presented in your study (Lines 235-236). Please explain this discrepancy and if you have done modifications to the data, how those were conducted.

** We thank the reviewer to point out the discrepancies between Wong et al. (2023) and our study. As the Global Ant Biodiversity database (GABI) is regularly updated (Guenard et al. 2017), the discrepancy in alien species distribution between our study and the one of Wong et al. comes from the last download of alien species distribution (respectively November 2020 and June 2022). However, after downloading the alien species distribution from Wong et al. 2022, we reran all our analysis and although they described more alien species, the discrepancies between the two datasets were not changing the main result of our study. We provide the final output compared maps below. We keep in the manuscript the results based on the version of 2020 as we had information on both non-alien and alien species at that date, and not only alien species.

All ants	Database 11/2020	Updated from Wong et al. 2023
Native range	[figure redacted]	[figure redacted]
Current range	[figure redacted]	[figure redacted]

Alien ants only	Database 11/2020	Updated from Wong et al. 2023
Native range	[figure redacted]	[figure redacted]
Current range	[figure redacted]	[figure redacted]

Minor comments

All over your text, you would need to refer to biogeographic realms and not “regions” as those refer to a different unit.

** We thank the reviewer for this comment on which we agree. We decided to refer to “biogeographic realms” or “realms” when talking about the broadest distributional patterns of ant species. We kept regions as a term used in our homogenization study, as we looked at finer scale homogenization degree (polygons within realms).

Line 18: the dispersal of alien species is of both accidental and deliberate nature, as thousands species of plants, vertebrates and invertebrates have been spread for economic reasons (agriculture, forestry, biological control, landscaping...).

** We agree. We added the aspect of deliberate dispersal which was previously unintentionally omitted.

Line 62-63: this sentence is factually incorrect as species of the *Formica rufa* group were introduced in Central Italy (see work by Pavan in the 1950's & 60's) and later on in Eastern Canada in the 70's.

** We thank the reviewer for this interesting remark. We decided to check out the status of this species on Antmaps (the online platform produced based on the GABI dataset). We noticed that *Formica rufa* is considered as native from Palearctic (comprising Italy). Even though it has been voluntarily introduced, it is not considered as an ant invasion (not introduced and established outside of its native range). Additionally, established population of *Formica rufa* in the Canada based on the same data source are dubious. So those records are not considered in our analysis. No ant invasions could therefore be considered as deliberate to our knowledge.

Line 93: a detail but as you did not use all valid ant species in your analyses, this number should be 1.8% (292/15850)

** The reviewer is right. We therefore decided to explain to what exactly those 2.1% were referring to (l.107).

Lines 180-183: see also Moser et al. (2018) which includes ants as well as other taxa.

** We added the reference.

Lines 190-192: I don't understand the point of this sentence as the authors acknowledge the benefit of increased spatial resolution but did not use the latest and most comprehensive data available. See also Kass et al. 2022 presenting a much finer spatial resolution and identification of undersampled regions.

** We thank the reviewer for the comment revealing a lack of clarity. In the paper of Kass et al. 2022, they used data downloaded from the GABI database on the 14th of July 2020, on which they applied spatial distribution modelling to infer potential gaps on ant distribution

knowledge. We downloaded the data in November 2020. The second difference with Kass et al. is that they used georeferenced occurrences points (finer scale than our polygon definition). However, as we were doing pairwise comparison of beta diversity between geographical unit, increasing the spatial resolution is for now very complicated because not all species have the same resolution in range delineation, and the occurrences point are not available for many species. Therefore, choosing a very fine scale could increase the bias in species richness by spatial unit, with the risk that under-sampling of some regions will more heavily bias the results due to the false absence. Also, our point in that sentence (l. 204), was to highlight that we studied realm patterns, but it could be interesting to have finer scale delineation of species assemblages: “Future studies may analyze biogeographic patterns at finer resolution^{44,47,48.}”

Lines 192-195: see Yang et al. 2021

Lines 195-197: See direct examples in Bernardo-Madrid et al. 2019.

**** As suggested by the reviewer, we added the two previous references.**

Lines 199: replace “severe” by “deep” as you did not measure any effects directly and local or regional extinctions consequences on native species unknown.

**** We modified that.**

References:

**** We added the following references suggested by the reviewer when they were not already cited.**

Bernardo-Madrid et al. 2019. Ecology Letters 22, 1297–1305

Capinha et al. 2015. Science 348, 1248–1251

Fricke & Svenning 2020. Nature 585, 74–78.

Holt et al. 2013. Science 339, 74.

Kass et al. 2022. Science Advances 8(31), eabp9908.

Liu et al. 2021. Curr. Zool. 67, 393–402.

Moser et al. 2018. PNAS 115, 9270–9275.

Yang et al. 2021. Nature Communications 12, 7290.

REVIEWER COMMENTS

Reviewer #1 (Remarks to the Author):

The authors have successfully addressed my comments and suggestions. I also congratulate them for producing such an interesting and relevant piece of work.

Reviewer #2 (Remarks to the Author):

Thanks very much for addressing all my comments. Although nearly all my concerns have been properly addressed, I have a few more that remain below, and they are purely textual. Two of these remaining concerns make the important point that this paper still needs to frame its results against biogeographic results for ants released by very recent and comprehensive studies, two of which use the same dataset as this study. Some discussion of how the current results compare with these past studies is needed. Currently they are not given sufficient explanation.

As for interpretation of the results, I am not an expert on ant biogeography so I cannot comment on their validity.

1. About my main concern regarding the sampling bias for alien ants compared to native ones, although it is true that the effect would not be as strong for polygon-level estimates compared to point-level or grid cell-level estimates, the effect is likely still there. If more data exists for alien species, then more polygons would be sampled, and undersampling would be more extreme for non-alien species. Thus, it would be a good idea to include at least a statement in the Discussion explaining how this could have affected results.

2. L83: I recommended before to include a reference to Kass et al. here. The reason for this is that it is the most comprehensive, fine-grained, and recent estimation of distributions for all described ant species. It also uses the GABI database, which you use here. Yet there is still no mention of this study or reference in the Intro, nor any explanation of what is currently understood about ant distributions globally. Here would be a good opportunity to

add a sentence explaining what Kass et al. found in their study using new distributional data on ants from the GABI database, and that this study will use this new database to explore biotic homogenization via ant invasions. In sum, it is important to set the baseline for what has been done to date, especially what is most current at the time of writing, and to set up the current study as building upon this base of knowledge.

3. L205: GABI not introduced in the text yet.

4. L212: Not sure why this sentence says "Future studies may analyze biogeographic patterns at finer resolution", yet references studies that already did this. Please rephrase this to emphasize that finer-scale studies have already been done, and explain what they found and how they compare to the results of this study. Wang et al. even uses a bioregionalization approach in a very similar way to the approach used in this study. It is responsible to thus give space to sufficiently explain these other studies that came before this one and how they compare to the current results.

Reviewer #3 (Remarks to the Author):

Lines 35-37: Add other studies showing changes in biogeographic boundaries as a consequence of species introduction in mammals, birds (Bernardo-Madrid et al. 2019)

Line 62: Formicidae should not be italicized. In taxonomy, only the genus and species (or subspecies) levels are italicized. Higher taxonomic levels, such as the family level (Formicidae) remain written normally.

Line 81: you did not include all described ant species (see below), thus modify

Lines 107: Modify so it reads "only 2.1% of ant species used in this study (13,758)" as over 14,141 species and 1755 subspecies (many actually representing valid species) are currently recognized, so a total of nearly 16,000 species and subspecies.

Line 205: GABI is just compiling records published within literature, it would thus be best to

refer directly to those authors contributing directly to the knowledge on the spread of exotic species.

Line 206-208: You should emphasize the high undersampling on islands as shown in Liu et al. 2023 as this may directly affect your conclusion about homogenization.

Lines 208-209: This sentence is a bit confusing as you refer only to newly described species, however, homogenization levels are impacted by both newly recorded and newly described species.

On figures 2 and 3: It seems that there is a different representation of India in function of the panel considered. For instance, on figure 2, India is divided into state level on panel A), B) and C) but not on D; similarly, India is divided into state level on panel A) but not on B). Did you use different levels to conduct the analyses?

We thank the reviewers for their additional constructive comments. We are confident that this second revision has further improved the manuscript. In particular, we have added some text to the discussion to discuss our findings in the light of recent biogeographic studies on ants. We also provide details concerning the source of the data used in our study. Please find below our detailed point-by-point replies (** in red) to the reviewer's comments (in black).

Additional reviewers' request

The data you are using for introduced species' ranges is not made available in the data release from GABI. Please can you clarify the source of these data and whether there was permission for their use, as well as how the data from Wong et al. 2023 were used.

** We thank the additional reviewers' request to point out that the description of our methods section was unclear. We have now clarified the source of both native species' ranges and introduced species' range. This data was publicly available on the Antmaps.org website which displays the species' ranges by sub-country level polygons and the distinction between the native and the introduced ranges of alien species. According to this website, researchers who use this data should cite two papers (Guénard et al. 2017, *Myrmecological News*; Janicki et al. 2016 *Ecological Informatics*), which we did. However, for clarification, we have also now cited the website itself.

Concerning the distribution of the 292 alien ant species in their native and introduced ranges, we additionally compared our data with a recent release of alien ant species distribution, available as supplementary material in the paper by Wong *et al.* 2023, and with the most recent ranges available on Antmaps.org. As we found less than 3% disagreement, we decided to use the November 2020 alien ant species ranges (both native and introduced) as we had the non-alien ant species distribution from the same year. We added this in the methods : "For alien ant species, we checked if native and introduced ranges were consistent with the recently published study on alien ant species by Wong *et al.* (2023)³⁷ as well as with the most recent distributions available on Antmaps.org (as of July 2023)." (l.274-275).

Also, in your manuscript, we would like you to improve your explanation and justification for how native and introduced ranges were assigned, including data cleaning processes that were involved and any limitations with the data.

** We added relevant information about data extraction, assignment, cleaning processes and limitations with the data in the paragraph "distributional data and pre-processing" in the Methods.

Reviewer #1 (Remarks to the Author):

The authors have successfully addressed my comments and suggestions. I also congratulate them for producing such an interesting and relevant piece of work.

** We thank Reviewer 1 for this compliment and previous suggestions and are really pleased that our work is considered interesting and relevant.

Reviewer #2 (Remarks to the Author):

Thanks very much for addressing all my comments. Although nearly all my concerns have been properly addressed, I have a few more that remain below, and they are purely textual. Two of these remaining concerns make the important point that this paper still needs to frame its results against biogeographic results for ants released by very recent and comprehensive studies, two of which use the same dataset as this study. Some discussion of how the current results compare with these past studies is needed. Currently they are not given sufficient explanation.

As for interpretation of the results, I am not an expert on ant biogeography so I cannot comment on their validity.

** We thank Reviewer 2 for the positive general comment and aimed at better discussing our results regarding previous studies on ant biogeography. We detail below what we did point-by-point.

1. About my main concern regarding the sampling bias for alien ants compared to native ones, although it is true that the effect would not be as strong for polygon-level estimates compared to point-level or grid cell-level estimates, the effect is likely still there. If more data exists for alien species, then more polygons would be sampled, and undersampling would be more extreme for non-alien species. Thus, it would be a good idea to include at least a statement in the Discussion explaining how this could have affected results.

** We agree with the comment and added a line of discussion on that point (3d paragraph of the Discussion).

2. L83: I recommended before to include a reference to Kass et al. here. The reason for this is that it is the most comprehensive, fine-grained, and recent estimation of distributions for all described ant species. It also uses the GABI database, which you use here. Yet there is still no mention of this study or reference in the Intro, nor any explanation of what is currently understood about ant distributions globally. Here would be a good opportunity to add a sentence explaining what Kass et al. found in their study using new distributional data on ants from the GABI database, and that this study will use this new database to explore biotic homogenization via ant invasions. In sum, it is important to set the baseline for what has been done to date, especially what is most current at the time of writing, and to set up the current study as building upon this base of knowledge.

** We thank the Reviewer for the very interesting comment. It is true that we did not mention Kass and colleagues' work in our Introduction and we changed this. We strongly agree that it is important to set the baseline for what has been done to date and to mention the studies that have been published recently.

3. L205: GABI not introduced in the text yet.

** We corrected this.

4. L212: Not sure why this sentence says "Future studies may analyze biogeographic patterns at finer resolution", yet references studies that already did this. Please rephrase this to emphasize that finer-scale studies have already been done and explain what they found and how they compare to the results of this study. Wang et al. even uses a bioregionalization approach in a very similar way to the approach used in this study. It is responsible to thus give space to sufficiently explain these other studies that came before this one and how they compare to the current results.

** We thank the reviewer for this comment that also refers to the previous one about mentioning the baseline and the most recent studies. We have now highlighted these most recent studies published on ants and compared them to our results:

"However, future studies on the impact of ant invasions may analyze biogeographic patterns at finer resolution⁵³ to detect more precisely biogeographic transitions, as for recent studies on bioregionalization in European ants⁵⁴ and global native ant biodiversity⁴²." (l.224-226).

Reviewer #3 (Remarks to the Author):

Lines 35-37: Add other studies showing changes in biogeographic boundaries as a consequence of species introduction in mammals, birds (Bernardo-Madrid et al. 2019)

** We clarified our first sentence and we modified accordingly to the comment. "Moreover, recent research on vertebrates has shown that human-mediated introductions and species extinctions alter biogeographic boundaries, with marked differences according to the taxonomic group¹⁵." (l.39-40).

Line 62: Formicidae should not be italicized. In taxonomy, only the genus and species (or subspecies) levels are italicized. Higher taxonomic levels, such as the family level (Formicidae) remain written normally.

** We thank the reviewer for the remark and modified accordingly.

Line 81: you did not include all described ant species (see below), thus modify

** We clarified the sentence as it was also suggested below (l.88-89).

"[...] all ant species with known distribution records (13,758 species, hereafter referred to as "all ant species") [...]"

Lines 107: Modify so it reads "only 2.1% of ant species used in this study (13,758)" as over 14,141 species and 1755 subspecies (many actually representing valid species) are currently recognized, so a total of nearly 16,000 species and subspecies.

** We agree and have modified this.

Line 205: GABI is just compiling records published within literature, it would thus be best to refer directly to those authors contributing directly to the knowledge on the spread of exotic species.

** We indeed agree with the reviewer and modified accordingly.

Line 206-208: You should emphasize the high undersampling on islands as shown in Liu et al. 2023 as this may directly affect your conclusion about homogenization.

** We clarified that point (l.228-233):

"We acknowledge that islands are largely under-explored with, for example, more than 108 large islands globally (with an area >200 km²) that have received no sampling effort⁵⁰. This under-sampling may have affected our estimate of homogenization on islands, although we believe that the general pattern of homogenization along the tropical belt is likely to be robust. The release of a new database of global ant biodiversity on islands⁵⁰ [...]"

Lines 208-209: This sentence is a bit confusing as you refer only to newly described species, however, homogenization levels are impacted by both newly recorded and newly described species.

** This is true, we detailed and clarified that point.

On figures 2 and 3: It seems that there is a different representation of India in function of the panel considered. For instance, on figure 2, India is divided into state level on panel A), B) and C) but not on D; similarly, India is divided into state level on panel A) but not on B). Did you use different levels to conduct the analyses?

** We thank the reviewer for pointing that mistake out. All analyses were conducted at the same level, using bentities (polygons of countries or state level). The problem just arose during the editing of the figures in Adobe Illustrator. We modified this.

REVIEWER COMMENTS

Reviewer #3 (Remarks to the Author):

This is now the third time that I have the opportunity to review the manuscript submitted by Aulus-Giacosa and collaborators.

I am satisfied by most of the responses that the authors provided to the comments regarding the changes or edits suggested by other reviewers and myself.

I must insist, however, for the authors to keep their analyses on the basis of the data available in Wong et al. (2023). While this may be a misunderstanding, the data regarding introduced records in GABI/antmaps are not freely usable, in contrast to those for native ranges released in 2020 (but see conditions for AntWeb records). Their extraction from the maps presented on antmaps.org is not allowed, nor encouraged, and thus the data used in your study were obtained without permission (not only from the main authors of GABI but also their collaborators). This dataset has been carefully collected, curated and with records then identified individually as representing different levels of introduction, to ultimately be released in Wong et al. (2023). In addition, the data presented on antmaps do not represent the final stage of data curation/knowledge and should thus not be used blindly (as recommended by their authors).

Minor suggestions:

Please correct “antmaps” not “Antmaps”

I also note that the mention of AntWeb, which I believe was present in a previous version has disappeared. However, as the records for native range provided by the dataset available on antmaps.org did not include the AntWeb records, you should clearly explain how these data were incorporated.

Dear Reviewers,

We would like to thank Reviewers 1 and 2 for their last revision of the paper that further improved the manuscript. We additionally thank Reviewer 3 for the additional constructive comments. We are confident that this third revision has solved the concerns regarding the use of the data. We have edited the manuscript (text, figures, supplementary figures, and all statistics) according to the suggestion of reviewer 3 to exclusively use most up-to-date dataset on alien species published by Wong *et al.* (2023). We provide details concerning the source of the data used in our study. Overall, the re-analysis based on solely the data by Wong and colleagues has had no impact on the conclusions presented in this manuscript. Please find below our detailed point-by-point replies (** in red) to the comments (in black).

Reviewer #3 (Remarks to the Author):

This is now the third time that I have the opportunity to review the manuscript submitted by Aulus-Giacosa and collaborators.

I am satisfied by most of the responses that the authors provided to the comments regarding the changes or edits suggested by other reviewers and myself.

I must insist, however, for the authors to keep their analyses on the basis of the data available in Wong *et al.* (2023). While this may be a misunderstanding, the data regarding introduced records in GABI/antmaps are not freely usable, in contrast to those for native ranges released in 2020 (but see conditions for AntWeb records). Their extraction from the maps presented on antmaps.org is not allowed, nor encouraged, and thus the data used in your study were obtained without permission (not only from the main authors of GABI but also their collaborators). This dataset has been carefully collected, curated and with records then identified individually as representing different levels of introduction, to ultimately be released in Wong *et al.* (2023). In addition, the data presented on antmaps do not represent the final stage of data curation/knowledge and should thus not be used blindly (as recommended by their authors).

**** We thank Reviewer 3 to point out that the most up-to-date curated data about alien species distribution was made publicly available in Wong *et al.* (2023). We therefore decided to rerun all analyses using this dataset for the 309 alien species established outdoors. Major findings remained unchanged, and all the small numerical changes can be tracked in red along the main text/figures/supplementary.**

Minor suggestions:

Please correct “antmaps” not “Antmaps”

**** We corrected.**

I also note that the mention of AntWeb, which I believe was present in a previous version has disappeared. However, as the records for native range provided by the dataset available

on antmaps.org did not include the AntWeb records, you should clearly explain how these data were incorporated.

** We modified in the Method section as well as the Data availability statement to clarify:
“The raw data that support this study were sourced from the webmaps displayed on antmaps.org which is linked to the Global Ant Biodiversity Informatics (GABI) database^{31,32} and Wong *et al.* (2023)³⁷.”

REVIEWERS' COMMENTS

Reviewer #3 was satisfied with the latest version and had no further comments.